# FAITHTHINKER: DIALECTICAL REASONING FOR NOISE-ROBUST LLMS

## ABSTRACT

Large Language Models (LLMs) have shown strong capabilities across a wide range of tasks. However, they remain vulnerable to noisy or adversarial contexts, often producing unfaithful or hallucinatory outputs. To address these weaknesses, recent work has integrated LLMs with Retrieval-Augmented Generation (RAG) and external tools. While effective, these approaches still suffer from error propagation, as existing structured reasoning methods cannot reliably detect and correct mistakes during intermediate steps. We propose FaithThinker, a reasoning framework designed to improve contextual faithfulness. At its core is Self-Questioning and Verification (SQV), a reasoning paradigm inspired by dialectical thinking. SQV allows models to question, verify, and revise intermediate reasoning steps in a single pass. To extend this capability, we introduce SQV-Alignment, an adversarial context–augmented fine-tuning method that efficiently transfers SQV from large to smaller LLMs. Experiments demonstrate that FaithThinker achieves state-of-the-art robustness under both clean and noisy conditions. SQV reduces hallucinations by up to **30.6%** compared with Chain-of-Thought, and generates reasoning paths **3×** shorter than iterative methods such as Self-Refine. These results highlight FaithThinker's ability to enhance contextual faithfulness, mitigate hallucinations, and improve efficiency in challenging environments.

## 1 INTRODUCTION

Large language models (LLMs) play a central role in modern AI systems, supporting a diverse range of applications from question answering to scientific research (DeepSeek-AI et al., 2025; Zhang et al., 2025; Wei et al., 2025). To overcome the limitations of their parametric knowledge, LLMs are increasingly combined with retrieval-augmented generation (RAG) to access dynamic and domain-specific information (Guu et al., 2020). While these integrations enhance model utility, they also exacerbate the challenge of faithfulness under noisy or adversarial contexts. Here, faithfulness refers to the ability of a model to remain consistent with external evidence and avoid producing hallucinations. Despite recent progress, current LLMs still struggle to maintain faithfulness under imperfect conditions. Contextual information retrieved from the Internet varies in quality and may contain irrelevant or misleading content, and in some cases, it can be maliciously manipulated. Recent studies have shown that in such noisy environments, LLM performance drops significantly, producing hallucinations at a much higher rate (Ming et al., 2025).

Structured reasoning has emerged as a promising approach to address this challenge. By decomposing complex tasks into interpretable steps and encouraging models to justify intermediate decisions through explicit reasoning chains, structured methods help reduce hallucinations. However, its effectiveness is inherently limited in scenarios involving missing, corrupted, or adversarial inputs, where even systematic reasoning frameworks cannot guarantee robustness (Turpin et al., 2023; Xiang et al., 2024; Xu et al., 2024). A more insidious challenge arises from error cascades: initial hallucinations can propagate through subsequent reasoning steps, progressively distorting the entire inference trajectory (Zhang et al., 2023; Feng et al., 2025). As displayed in Figure 1, we categorize hallucinations in structured reasoning for context-dependent tasks into two types: (i) *Input-to-Trajectory Hallucinations*: hallucinations that arise from inconsistencies between the reasoning trajectory and the input context; and (ii) *Intra-Trajectory Hallucinations*: errors that occur within the reasoning trajectory itself. Unlike factual hallucinations, which can often be mitigated by introducing external information(Li et al., 2024; Song et al., 2024), faithfulness hallucinations in structured

reasoning are deeply embedded in the model's internal inference mechanisms. As such, they present a unique challenge for reliable reasoning in LLMs and highlights a core challenge: **achieving robust reasoning under noisy or adversarial contexts remains unresolved**.

To address these limitations, we propose *FaithThinker*, a reasoning framework designed to enhance contextual faithfulness under noisy or adversarial conditions. At the framework's core is *Self-Questioning and Verification (SQV)*, a reasoning paradigm inspired by dialectical thinking. Human reasoning often progresses by actively questioning assumptions, considering opposing perspectives, and refining conclusions. SQV operationalizes this process within LLMs: at each intermediate step, the model explicitly questions, verifies, and corrects its own outputs. This single-pass mechanism mitigates error propagation and produces more robust reasoning trajectories without the overhead of iterative refinement. We further extend SQV through *SQV-Alignment*, a method

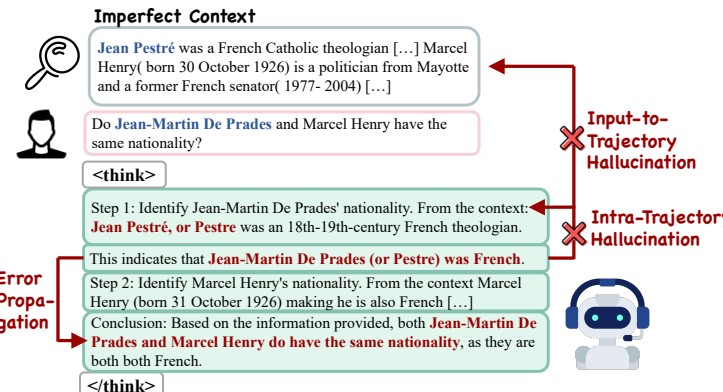

Figure 1: When the retrieved context contains inaccuracies (e.g., malicious name substitutions), the LLM is prone to hallucinations, including Input-to-Trajectory Hallucination and Intra-Trajectory Hallucination. Furthermore, the LLM propagates this error, producing an incorrect final output.

for transferring SQV capabilities from large to small LLMs. SQV-Alignment employs adversarial context–augmented fine-tuning, where training data is constructed to include noisy and misleading retrievals. This design equips smaller LLMs with zero-shot SQV reasoning skills, making FaithThinker efficient and widely applicable in practical settings where computational resources are limited.

We conduct extensive evaluations across six benchmark datasets and three distinct model families. Results show that FaithThinker consistently achieves state-of-the-art robustness in both clean and noisy conditions. Specifically, compared to vanilla CoT, SQV reduces hallucination rates by up to 24.1%, 19.0%, and 30.6% for Qwen2.5-3B-Instruct (Team, 2024), LLaMA3.1-8B-Instruct (Grattafiori et al., 2024) and Deepseek-R1 (DeepSeek-AI et al., 2025), respectively. At the same time, SQV demonstrates superior reasoning efficiency: it achieves a 4× shorter reasoning path than self-refine—which relies on iterative feedback—while attaining 10% higher average accuracy, confirming its ability to produce more precise reasoning trajectories. These comprehensive results validate FaithThinker's effectiveness in enhancing contextual faithfulness, mitigating hallucinations, and maintaining robust reasoning performance even in imperfect contexts, thereby addressing key limitations of existing structured reasoning approaches. The contributions of this work can be summarized as follows:

- **Problem characterization:** We investigate the faithfulness challenge in reasoning methods operating in noisy contexts, identifying two distinct types of hallucinations that undermine contextual faithfulness in reasoning models: *Input-to-Trajectory Hallucinations* and (ii) *Intra-Trajectory Hallucinations*.

- **Methodology:** We introduce *FaithThinker*, a reasoning framework that embeds *dialectical thinking* into LLMs. It features a structured reasoning paradigm named SQV that enables LLMs to iteratively hypothesize, question, verify, and refine intermediate steps within a single pass. We also provide SQV-Alignment that efficiently transfers SQV capabilities from large teacher models to smaller student models, thereby narrowing the performance gap under resource constraints.

- **Comprehensive validation:** We conduct extensive experiments across a wide range of benchmarks, and results show that FaithThinker consistently reduces hallucination rates (up to **30.6%** lower than Chain-of-Thought) and achieves substantially shorter reasoning trajectories (up to **4×** more concise than iterative refinement methods) across model families. Furthermore, we provide theoretical insights into SQV's superiority in Appendix A.

## 2 RELATED WORK

### 2.1 FAITHFULNESS HALLUCINATION

Hallucination in LLMs refers to confident yet incorrect outputs, manifesting either as factual errors (contradicting real-world knowledge) or faithfulness failures (deviating from the provided context) (Huang et al., 2025; Ye et al., 2023). This issue is particularly critical in tasks such as RAG and contextual Question Answering (QA), where maintaining faithfulness to the context is essential for ensuring the reliability and trustworthiness of the generated content (Zhou et al., 2024; Ming et al., 2025). Recent advances in RAG frameworks have aimed to improve answer reliability. However, the variability in source quality—such as noisy or contradictory evidence—presents a significant challenge in maintaining faithfulness in LLMs, especially when multi-hop reasoning chains are involved. Studies have shown that LLMs, even when fluent and seemingly coherent, often suffer from faithfulness hallucination, particularly under noisy contexts (Du et al., 2024; Turpin et al., 2023; Ming et al., 2025). These findings highlight the need for robust methods that ensure faithful generation in real-world conditions, where knowledge retrieval is imperfect, and intentional adversarial manipulation is possible.

### 2.2 STRUCTURED REASONING PATTERNS FOR LLMS

Traditional methods for mitigating hallucinations, such as external knowledge verification or post-hoc filtering, only correct errors after text generation, without improving the model's internal reasoning capabilities (Bi et al., 2024). As language modeling has evolved beyond simple next-word prediction, more research has focused on enhancing LLMs' reasoning capabilities to reduce hallucinations (Dhuliawala et al., 2023; Yu et al., 2024). Prompting techniques have played a key role in advancing the reasoning capacities of LLMs. Among these, Chain-of-Thought prompting (Wei et al., 2023) has emerged as a leading paradigm, inspiring numerous variants such as Least-to-Most (Zhou et al., 2023) and Auto-CoT (Zhang et al., 2022). These methods share a common "divide and conquer" strategy: guiding LLMs to break down complex problems into manageable sub-tasks, solve each step systematically while documenting the reasoning process, and finally synthesize a coherent answer. To further advance reasoning performance, current methods can be broadly categorized into the following two paradigms:

- **Single-step Extension**: A primary mechanism in single-step extension involves selecting or voting for the best reasoning path from multiple independently sampled CoT trajectories. Techniques such as Best-of-N sampling (Cobbe et al., 2021) and Self-Consistency (Wang et al., 2023) exemplify this approach. Further innovations like Tree of Thoughts (ToT) (Yao et al., 2023) and Graph of Thoughts (GoT) (Besta et al., 2024) introduce search-based schemes to expand the scope and depth of reasoning path exploration. However, these methods often require predefining a fixed candidate size for each node, which can result in either redundancy or insufficiency.

- **Iterative Optimization-based Extension**: These methods enhance reasoning by allowing models to iteratively critique and revise their own answers. Approaches like Self-Check (Miao et al., 2023) and Self-Refine (Madaan et al., 2023) prompt models to detect and correct errors using only internal knowledge. However, self-correction is often limited in reasoning-intensive tasks, where models struggle to identify and rectify errors without external feedback. Moreover, iterative methods can introduce new mistakes, reducing overall accuracy (Huang et al., 2024). Additionally, these approaches are computationally inefficient due to the need for multiple reasoning passes.

In summary, neither single-step extension nor iterative optimization-based reasoning methods significantly improve the model's intrinsic reasoning capacity. While these methods may reduce hallucinations in some contexts, they fall short in ensuring robust, self-correcting reasoning under real-world conditions, such as imperfect knowledge retrieval and intentional adversarial manipulation. This highlights the urgent need for more effective methods that can improve faithfulness and reasoning robustness in challenging environments.

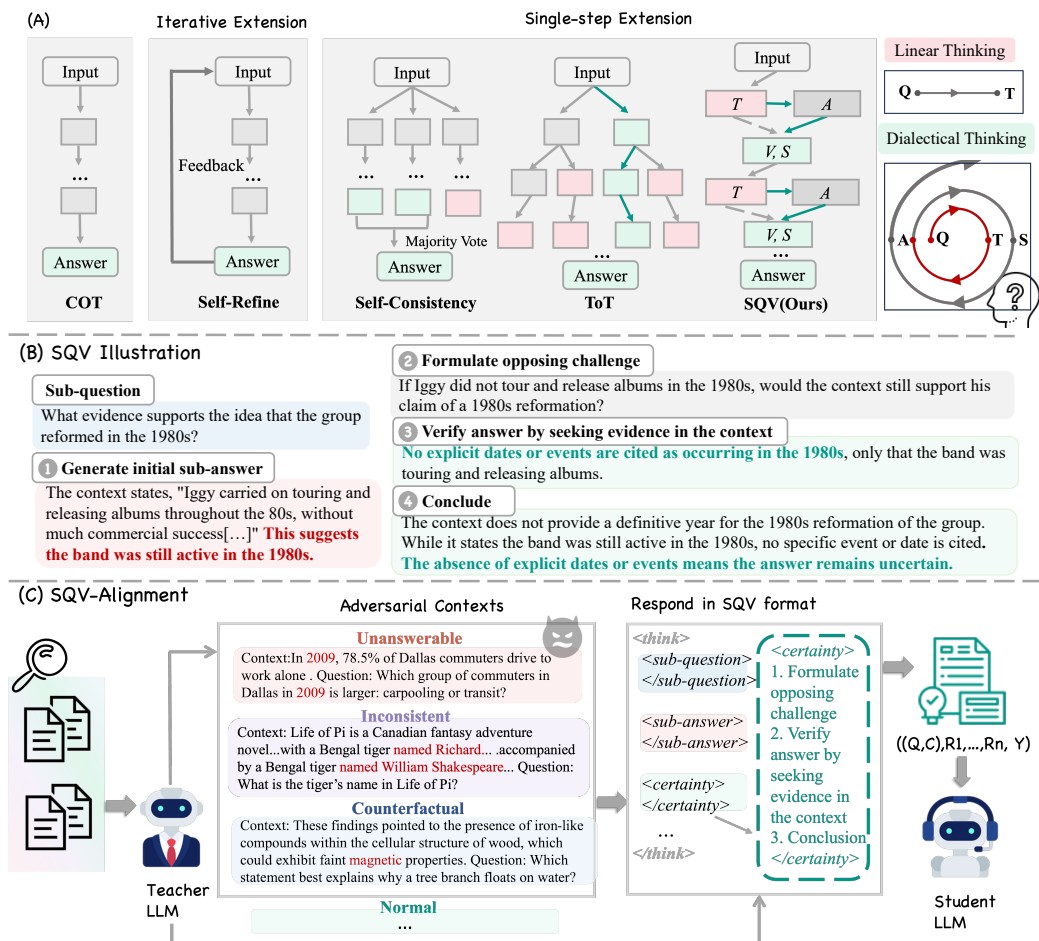

Figure 2: Illustration of the proposed FaithThinker. (A) Comparison of reasoning patterns. Each rectangular shape symbolizes a distinct thought. Unlike previous study, SQV integrates the verification process within each atomic reasoning step, enabling iterative refinement within a single forward pass. $T$, $A$, $V$ and $S$ represent Thesis, Antithesis, Verification and Synthesis respectively. (B) SQV integrates the dialectical process into four key stage, employing self-questioning and iterative verification to refine initial responses and thus achieve robust reasoning. (C) SQV-Alignment that systematically transfers SQV reasoning proficiency from large teacher LLMs to small student LLMs integrated with adversarial samples.

## 3 METHOD

### 3.1 HALLUCINATION IN CONTEXTUAL REASONING

In context-dependent tasks, the correct prediction or output depends not only on the input query but also on additional contextual information, which provides essential supporting details (Fan et al., 2024; Bi et al., 2024). Formally, let $Q$ be the input query, $C$ be the context, and $Y$ be the target output. A context-dependent task is defined as modeling the conditional probability $P(Y \mid Q, C)$. When integrated with reasoning, the generation process can be formalized by introducing a latent variable $R$, representing the reasoning trajectory. Consequently, the probability $P(Y \mid Q, C)$ in contextual reasoning is then marginalized over all possible $R$:

$$P(Y \mid Q, C) = \int P(Y \mid R, Q, C) \times P(R \mid Q, C) dR. \tag{1}$$

This formalization allows us to capture the dependencies between the input, context, reasoning process, and final output, which are crucial for ensuring contextual faithfulness in tasks that involve complex reasoning chains. While prior research on hallucinations in context-dependent tasks has

primarily focused on hallucinations in the final output (Ming et al., 2025; Huang et al., 2025), we extend this investigation to hallucinations within the reasoning process itself. Through failure case studies (As displayed in Figure 1), we observed that hallucinations within intermediate reasoning steps can lead to error cascades, where an early mistake propagates through the reasoning process, compounding the final output's inaccuracy.

We consider two core types of hallucinations in contextual reasoning. **Input-to-Trajectory Hallucination**: This type refers to reasoning steps that are inaccurate or unsupported by the input, leading to inconsistencies between the context and intermediate reasoning steps. **Intra-Trajectory Hallucination**: This refers to logical inconsistencies within reasoning trajectories, where subsequent reasoning steps contradict each other. Formally, given $C$ as the input context, and $R = [r_1, r_2, ...r_n]$ as the reasoning trajectory, let Consistent$(a, b)$ represent the binary function indicating logical alignment between elements $a$ and $b$, we define Input-to-Trajectory Hallucination and Intra-Trajectory Hallucination as follows:

$$H_{C \to R} = 1(\exists r_k \in R \text{ where Consistent}(r_k, C) = 0), \tag{2}$$

$$H_{R \to R} = 1(\exists (r_i, r_j) \in R \text{ where Consistent}(r_i, r_j) = 0). \tag{3}$$

The Equation 2 captures the case where a reasoning step $r_k$ is inconsistent with the input context $C$, while the Equation 3 defines the situation where two reasoning steps $r_i$ and $r_j$ are logically inconsistent with each other. Unlike factual hallucinations that can often be corrected through external information, faithfulness hallucinations in structured reasoning stem from fundamental flaws in the model's internal inference processes. These errors pose unique challenges for achieving reliable reasoning in LLMs, as they are intrinsically tied to the model's reasoning mechanisms rather than simple knowledge gaps.

### 3.2 REASONING PATTERN

To address the challenges of faithfulness hallucination and error propagation in context-dependent reasoning, we propose FaithThinker, a novel reasoning framework that enforces self-questioning and verification at each intermediate step of the reasoning process. FaithThinker ensures that the model questions and verifies its reasoning in real-time, preventing errors from propagating and improving contextual faithfulness throughout the reasoning chain. By adopting this approach, FaithThinker enhances the model's ability to reason faithfully in noisy, context-dependent scenarios.

At the heart of FaithThinker is the Self Questioning and Verification (SQV) reasoning pattern. Drawing inspiration from how humans develop a deeper understanding through the interaction of contradictions and opposing viewpoints in learning, problem-solving, and decision-making (Basseches, 1984; King & Kitchener, 2012), SQV integrates the dialectical process—commonly understood as the "thesis-antithesis-synthesis" dynamic—into four key stages, as illustrated in Figure 2. This dialectical process mirrors the way humans engage with complexity by synthesizing evidence from diverse perspectives, avoiding overly simplistic black-and-white reasoning, and refining understanding through critical self-reflection. Specially, the SQV reasoning pattern involves four key stages:

**Thesis** : The model generates an initial sub-question decomposition and answers, aligning with standard Chain-of-Thought reasoning.

**Antithesis** : For each derived sub-answer, the model engages in critical self-interrogation by formulating opposing challenges (objections) to test the robustness of its initial reasoning.

**Verification** : The model re-examines the contextual information underlying each challenge, analyzing relevant evidence to either support or refute the original sub-answer.

**Synthesis** : Drawing from the insights generated in the previous steps, the model refines its initial answers, arriving at a well-validated conclusion.

This process closely follows the thesis-antithesis-synthesis framework, which mimics how human cognition refines beliefs by integrating opposing views and reconciling contradictions. Formally, we define the reasoning process in terms of the following joint probability, where the model evaluates each stage of reasoning within the context of the input query $Q$, context $C$, and the generated reasoning elements $T$ (initial answer), $A$ (objection), $V$ (verification), and $S$ (synthesis):

$$P(Y \mid Q, C) = \iiint \underbrace{P(Y \mid S, V, A, T, Q, C)}_{} \cdot P(V \mid A, T, Q, C) \cdot \underbrace{P(A \mid T, Q, C)}_{\text{Antithesis}} \cdot \underbrace{P(T \mid Q, C)}_{\text{Thesis}} \, dT dA dV. \tag{4}$$

$$\underbrace{\phantom{P(Y \mid S, V, A, T, Q, C) \cdot P(V \mid A, T, Q, C)}}_{\text{Synthesis based on evidence}}$$

This formalization represents the multi-stage reasoning process where the model synthesizes evidence, critiques its initial answers, verifies the validity of intermediate reasoning steps, and arrives at a refined conclusion. We additionally provide theoretical insights at Appendix A.

As in Figure 2, unlike conventional iterative optimization approaches, which require multiple rounds of answer generation and external verification, SQV integrates the verification process within each atomic reasoning step. This enables iterative refinement within a single forward pass, decoupling logical depth from computational overhead. By directly embedding antithesis generation and evidence-based validation into each sub-question resolution, SQV effectively circumvents the redundancy of token generation found in methods like Self-Refine, while mitigating the risk of error accumulation.

To guide the LLMs reasoning based on the SQV pattern, we use meta-prompts formatted with XML tags, which enhances both the interpretability and reasoning quality of the generated content (Caruccio et al., 2024; Yamauchi et al., 2023). These meta-prompts, shown in the Appendix B.1, instruct the model to reason in alignment with the four stages of SQV, ensuring a structured and systematic approach to contextual reasoning.

### 3.3 ADVERSARIAL TRAINING

We formally define a model's capacity to perform zero-shot SQV-format reasoning directly from meta-prompts as $SQV_{Zero}$. Experimental analysis reveals that while larger language models (e.g., Deepseek-R1) demonstrate robust $SQV_{Zero}$ capabilities, smaller models exhibit significant performance degradation in noisy environments, failing to achieve authentic SQV reasoning (see Appendix C.1 for detailed analysis). To address this capability gap, we develop an adversarial training procedure called **SQV-Alignment** that systematically transfers SQV reasoning proficiency from large teacher LLMs to small student LLMs. We subsequently define the enhanced reasoning ability achieved through this alignment process as $SQV_{Alignment}$.

#### 3.3.1 DATA COLLECTION

To equip smaller reasoning models with the ability to perform SQV reasoning, acquiring appropriate training data is crucial. Since manually annotating reasoning data is resource-intensive, we leverage Deepseek-R1, a state-of-the-art large reasoning model, to generate this data. This approach is both cost-effective and enhances reproducibility, enabling us to systematically collect large-scale training datasets. Our synthetic data generation process addresses both the input and output aspects of SQV reasoning (the pseudo-code of the automated pipeline provided in Appendix B.2).

**Regarding Input**: To enhance the noise robustness of LLMs, we incorporate noisy contexts into the fine-tuning data. Specifically, we follow the method in (Ming et al., 2025) and employ prompt instructions to guide Deepseek-R1 in generating three types of noisy text: Unanswerable Context, Inconsistent Context, and Counterfactual Context (Appendix D.1.1). We then combine these noisy contexts with randomly selected clean samples from SQuAD to construct the final input contexts used for training. **Regarding Output**: For the output generation, Deepseek-R1 is prompted with specific instructions (detailed in Appendix B.1) to produce reasoning traces that follow the SQV reasoning pattern.

#### 3.3.2 SUPERVISED FINE-TUNING PROCEDURE

During supervised fine-tuning (SFT), the model learns to generate a structured SQV reasoning sequence $S_{SQV} = (r_1, r_2, ..., r_n, y)$, where each $r_i = (h, q, e, c)$ represents a reasoning trace. Here, $h$ denotes the hypothesis, $q$ is the opposing question, $e$ is the evidence, and $c$ is the conclusion. We use the negative log-likelihood loss function for training: $\mathcal{L}_{SFT} = -\sum_{t=1}^{|S|} \log P(s_t \mid s_{<t}, X)$, where $s_t$ represents the $t$-th token in the ground-truth sequence $S$, and $s_{<t}$ refers to all tokens before position $t$. For computational efficiency, we employ Low-Rank Adaptation (LoRA) (Hu et al., 2021) during fine-tuning. This method updates only a small number of low-rank parameters inserted into each layer of the model, while keeping the original pretrained LLM parameters frozen. Through this fine-tuning procedure, SQV-Alignment trains the model to generate reasoning trajectories that follow the SQV pattern, thereby improving the contextual faithfulness of the reasoning process and mitigating hallucinations. This enables the model to perform structured reasoning steps, validate intermediate outputs, and ensure more reliable conclusions.

Critically, our goal is to endow the model with the ability to generate reasoning trajectories consistent with the SQV pattern. While numerous studies have explored integrating Reinforcement Learning (RL) to enhance reasoning abilities, our experiments reveal two key limitations with RL-based methods: (1) RL applied after SFT frequently degrades the model's ability to produce SQV-aligned reasoning traces. (2) RL-based methods consistently underperformed compared to the supervised fine-tuning paradigm across both noisy and standard benchmark datasets. Further analysis on this can be found in Appendix C.2. As a result, SFT serves as a more efficient and resource-conserving method for aligning the model's reasoning capabilities with the SQV pattern.

## 4 EXPERIMENTS

### 4.1 EXPERIMENT SETUP

**Models and Baselines.** We evaluate our method across three scales of reasoning models: two small LLMs—Qwen2.5-3B-Instruct (Team, 2024) and LLaMA3.1-8B-Instruct (Grattafiori et al., 2024)—and one large-scale LLM, Deepseek-R1 (DeepSeek-AI et al., 2025). For brevity, we adopt the abbreviated forms Qwen2.5-3B and LLaMA3.1-8B in subsequent discussions; Additionally, we compare our SQV pattern against three categories of baselines: 1) Vanilla Chain-of-Thought (COT)(Wei et al., 2023) reasoning. 2) Iterative Extension Method: Self-Refine(Madaan et al., 2023). 3)Single-step Extension Method: Self-Consistency(Wang et al., 2023) and Tree of Thoughts (ToT)(Yao et al., 2023). Detailed descriptions of each baseline are provided in Appendix D.2.

**Datasets and Implementation Details.** To enable structured reasoning with minimal training overhead, we prioritize data-efficient format alignment. For noise-robust reasoning model development, we curate a balanced 1K dataset by sampling 250 instances each from four distinct context categories: Unanswerable, Inconsistent, Counterfactual, and Normal scenarios, derived from SQuAD(Rajpurkar et al., 2016). We evaluate model robustness in both noisy and normal contextual environments: adversarial testing against noisy contexts using three datasets of FaithEval(Ming et al., 2025) and ConFiQA(Bi et al., 2024); to ensure model adaptability and assess generalization to out-of-domain (OOD) data, we also evaluate its performance on additional open-domain datasets including TriviaQA(Joshi et al., 2017) and Natural QA(Kwiatkowski et al., 2019). More implementation details can be found in the Appendix D.

**Evaluation Metrics.** We adopt two primary metrics: 1) Accuracy (Acc), which measures whether the final generations of the model align with the ground-truth(Asai et al., 2023). 2) Hallucination Rate (Hal), which measures whether the final answer fail to align with the context(Bao et al., 2024). More details can be found in the Appendix E.

### 4.2 MAIN RESULT

Table 1 illustrates the efficacy of our SQV method compared to the baselines in terms of Accuracy and Hallucination Rate. Notably, SQV achieves the best overall results on all three backbones compared to other reasoning methods. First, SQV exhibits substantial improvements over alternative reasoning strategies. It attains an average accuracy of 86.2% on Deepseek-R1, surpassing COT (69.9%) and Self-Consistency (73.8%) by 16.3% and 12.4%, respectively. Concurrently, SQV demonstrates significant hallucination reduction efficacy compared to vanilla CoT, achieving average relative decreases of 10.7% (Qwen2.5-3B), 11.2% (LLaMA3.1-8B), and 16.3% (Deepseek-R1) across evaluated models, confirming its superior capacity for contextual faithfulness preservation. Second, while methods like Self-Refine degrade performance compared to vanilla COT in noisy settings—for instance, reducing Qwen2.5-3B's performance to 56.6% on unanswerable scenarios (vs. 96.7% with SQV)—our approach maintains consistent hallucination suppression across models and noisy scenarios. We further extend our experiments to Qwen2.5-32B-Instruct and Qwen2.5-72B-Instruct using $SQV_{Zero}$ to investigate the scalability of the SQV reasoning pattern in Appendix D.4.

The performance improvement is consistent across all datasets, highlighting both the effectiveness and the strong generalization of our approach. **These results demonstrate that our self questioning and verification strategy can better guide the reasoning process and mitigate faithfulness hallucination persisting in LLMs, especially in challenging scenarios where baseline methods exhibit instability**.

Table 1: Comparative performance (%) of different reasoning methods under noisy and normal scenarios. The best results achieved using the same backbone model are highlighted in **boldface**. Cells significantly better than CoT are colored blue (deeper for greater improvement). Cells highlighted in purple represent the reduced hallucination rate of SQV compared to CoT.

| Method | Noisy Context | | | | Normal Context | |
|---|---|---|---|---|---|---|
| | FaithEval-U | FaithEval-I | FaithEval-C | ConFiQA | TriviaQA | NQ |
| **Qwen2.5-3B** | | | | | | |
| CoT | 88.9±1.2 | 98.2±0.5 | 60.2±1.5 | 62.6±1.4 | 62.0±1.6 | 49.6±1.3 |
| Self-Consistency | 80.8±1.1 | 84.8±1.3 | 65.3±1.4 | 73.5±1.0 | 76.3±1.2 | 49.0±1.5 |
| ToT | 48.7±2.0 | 72.0±1.8 | 52.7±1.7 | 63.2±1.3 | 72.5±1.1 | 49.8±1.6 |
| Self-Refine | 56.6±1.8 | 83.4±1.4 | 49.8±1.9 | 59.0±1.5 | 82.3±0.9 | 53.6±1.4 |
| SQV$_{Alignment}$(Ours) | **96.7±0.3** | **98.6±0.2** | **69.7±0.4** | **79.0±0.3** | **86.1±0.3** | **55.8±0.5** |
| Δ Compared to Vanilla C. | ↑ 7.8 | ↑ 0.4 | ↑ 9.5 | ↑ 16.4 | ↑ 24.1 | ↑ 6.2 |
| **LLaMA3.1-8B** | | | | | | |
| CoT | 72.4±1.3 | 95.5±0.6 | 48.4±1.6 | 57.4±1.5 | 68.4±1.4 | 52.8±1.2 |
| Self-Consistency | 80.6±1.2 | 97.2±0.5 | 63.0±1.1 | 66.4±1.2 | 77.5±1.0 | 55.4±1.3 |
| ToT | 34.2±2.2 | 44.0±2.0 | 51.7±1.8 | 44.0±1.7 | 63.7±1.5 | 47.0±1.6 |
| Self-Refine | 81.0±1.0 | 97.1±0.6 | 56.3±1.3 | 69.4±1.1 | 78.8±0.9 | 52.8±1.4 |
| SQV$_{Alignment}$(Ours) | **82.6±0.4** | **97.6±0.3** | **63.3±0.3** | **76.4±0.4** | **82.9±0.3** | **59.3±0.6** |
| Δ Compared to Vanilla COT. | ↑ 10.2 | ↑ 2.1 | ↑ 14.9 | ↑ 19.0 | ↑ 14.5 | ↑ 6.5 |
| **Deepseek-R1** | | | | | | |
| CoT | 65.0±1.4 | 99.3±0.3 | 53.7±1.5 | 72.6±1.3 | 70.9±1.3 | 58.2±1.3 |
| Self-Consistency | 65.6±1.4 | 99.3±0.3 | 59.8±1.4 | 79.2±1.1 | 76.8±1.2 | 62.1±1.2 |
| ToT | 76.0±1.3 | 93.5±1.2 | 48.6±1.7 | 54.2±1.6 | 46.3±1.8 | 46.6±1.7 |
| Self-Refine | 89.6±0.8 | 97.3±0.7 | 46.2±1.8 | 80.7±1.0 | 63.1±1.5 | 55.2±1.5 |
| SQV$_{Zero}$(Ours) | **95.6±0.2** | **99.8±0.1** | **75.1±0.3** | **93.6±0.2** | **86.3±0.3** | **66.8±0.4** |
| Δ Compared to Vanilla COT. | ↑ 30.6 | ↑ 0.5 | ↑ 21.4 | ↑ 21.0 | ↑ 15.4 | ↑ 8.6 |

## 4.3 ANALYSIS

**SQV Enhances the Quality of Reasoning Trajectories**   Beyond conventional performance metrics, we extend our analysis to evaluate the quality of reasoning trajectories. To systematically quantify reasoning trajectory quality, we introduce the **Top 10% Group Entropy** metric (Appendix E) inspired by (Fu et al., 2025). Our empirical evaluation compares reasoning trajectories generated by the two open-source models (Qwen2.5-3B and LLaMA3.1-8B) on the FaithEval-U dataset. As illustrated in Figure 3, SQV demonstrates consistently lower Top 10% Group Entropy compared to standard CoT across both architectures, indicating significantly decreased uncertainty and enhanced reasoning stability. Notably, the observed differences between CoT and SQV trajectories achieve statistical significance, demonstrating SQV's superiority in generating high-fidelity reasoning paths.

**SQV Achieves Superior Reasoning Efficiency.**   We further investigate the relationship between reasoning efficiency and accuracy across different reasoning patterns, using Deepseek-R1 as the experimental model. Here, Token Efficiency serves as our efficiency metric (Appendix E). As illustrated in Figure 5, baseline methods—including Self-Consistency, Self-Refine, and ToT —achieve moderate accuracy improvements but incur significant computational overhead and prolonged reasoning time due to multi-step sampling or iterative answer refinement (Refer to Table 5 for the specific numerical values). In contrast, SQV generates more concise reasoning paths, achieving state-of-the-art performance in both efficiency and accuracy. Specifically, SQV demonstrates 3× higher reasoning efficiency than Self-Refine while attaining a 10% higher average accuracy, validating the superiority of the SQV inference paradigm.

## 4.4 ABLATION

**Training Data Bias.**   To further investigate potential biases introduced by training data, we conduct training without SQV-formatted reasoning trajectories on Qwen2.5-3B using the same dataset (implementation details in Appendix D.5). As shown in Table 2, the influence of training data bias is minimal: while post-training performance improves on the datasets, gains remain modest compared to SQV-aligned training. Moreover, without SQV reasoning trajectories, the model exhibits limited generalization, with performance dropped significantly on TriviaQA, NQ and counterfactual tasks (For example, accuracy on TriviaQA drops from 86.1% to 52.4%). This stark contrast underscores

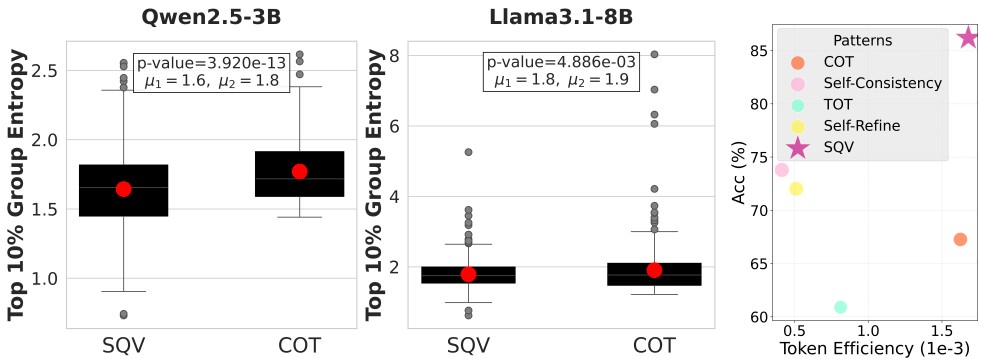

Figure 3: Top 10% Group Entropy Comparison for Qwen2.5-3B and LLaMA3.1-8B on FaithEval-U.

Figure 5: Token Efficiency vs. Accuracy.

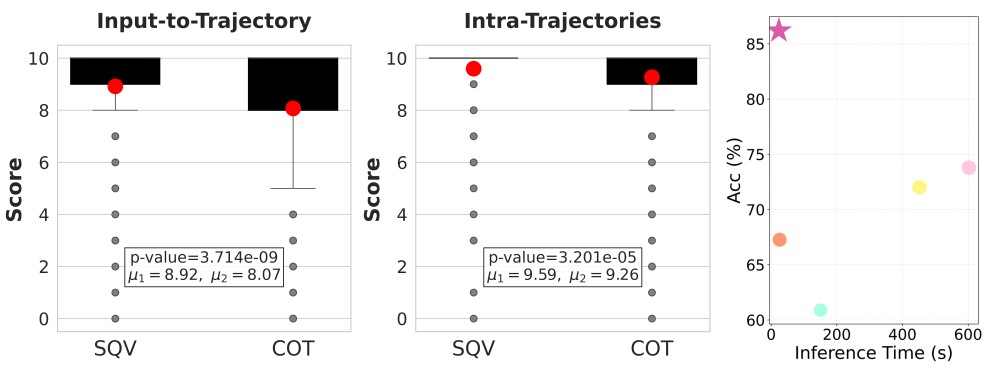

Figure 4: Hallucination Score Comparison for Deepseek-R1 on FaithEval-U.

Figure 6: Inference Time vs. Accuracy Across Reasoning Patterns.

Table 2: Results (%) of ablation study. The reported results are Acc.

| Method | Noisy Context | | | | Normal Context | |
| | FaithEval-U | FaithEval-I | FaithEval-C | ConFiQA | TriviaQA | NQ |
|---|---|---|---|---|---|---|
| **Qwen2.5-3B** | | | | | | |
| SQV$_{\text{Alignment}}$ | 96.7 | 98.6 | 69.7 | 79.0 | 86.1 | 55.8 |
| -w/o. SQV Format Alignment | 91.3 (-5.4) | 98.3 (-0.3) | 37.7 (-32.0) | 54.0 (-25.0) | 52.4 (-33.7) | 39.0 (-16.8) |
| -w/o. Dialectical Components | 81.0 (-15.7) | 74.5 (-24.1) | 37.7 (-32.0) | 53.4 (-25.6) | 57.9 (-28.2) | 41.8 (-14.0) |
| **Deepseek-R1** | | | | | | |
| SQV$_{\text{Zero}}$ | 95.6 | 99.8 | 75.1 | 93.6 | 86.3 | 66.8 |
| -w/o. Dialectical Components | 75.6 (-20.0) | 99.2 (-0.6) | 48.4 (-26.7) | 66.4 (-27.2) | 68.4 (-17.9) | 55.2 (-11.6) |

that SQV capability remains the primary driver of performance gains, with format alignment playing a more critical role than dataset-specific optimization.

**Dialectical Components.** To further validate the effectiveness of applying thesis-antithesis-synthesis reasoning at each atomic processing step, we conducted experiments on Qwen2.5-3B and Deepseek-R1 by replacing the component in the SQV with simplified verification (implementation details in Appendix D.5). As demonstrated in Table 2, the absence of the dialectical components resulted in consistent performance significant degradation across all evaluated datasets, with Qwen2.5-3B and Deepseek-R1 experiencing a 23.3% and 17.3% decrease in average accuracy respectively. This empirical evidence substantiates that dialectical thinking patterns critically enhance the model's ability to maintain contextual coherence during the reasoning processes.

## 5 CONCLUSION

While modern LLMs leverage RAG and external tools to access dynamic, domain-specific knowledge, their reliability diminishes in noisy or adversarial environments. Hallucinations proliferate when retrieved contexts are incomplete, misleading, or manipulated, as highlighted by recent studies(Ming et al., 2025). Structured reasoning methods mitigate this issue by decomposing tasks into interpretable steps but fail to ensure robustness when information is corrupted, leading to cascading errors in reasoning chains. To bridge these gaps, we propose FaithThinker, a framework that integrates Self-Questioning and Verification (SQV)—a novel reasoning paradigm inspired by human dialectical thinking. SQV embeds a single-pass verification mechanism where LLMs actively question, verify, and refine their intermediate outputs, minimizing error propagation without iterative refinement. FaithThinker further introduces SQV-Alignment, enabling knowledge transfer from large to small LLMs via adversarial context–augmented fine-tuning. Empirical evaluations across six benchmarks and three model families validate FaithThinker's robustness in noisy environments and superior efficiency gains. By addressing the limitations of existing structured reasoning methods, FaithThinker advances the deployment of reliable, contextually grounded LLMs in real-world, imperfect environments.

REPRODUCIBILITY STATEMENT

To ensure the reproducibility of our work, we have taken the following measures: (1) All implementation details, including model architectures, evaluation datasets, hyperparameters, and training procedures, are comprehensively documented in Section 4.1 and Appendix D; (2) The synthetic dataset generation process is fully specified in Section 3.3.1 with additional pseudo-code provided in Appendix B.2; (3) Our downloadable source code has been submitted as supplementary material.

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

# A  THEORETICAL INSIGHTS

The goal in this section is to **provide an intuitive perspective** towards the strength of SQV reasoning format instead of a rigorous proof. In noisy environments, the context $C$ may contain erroneous or misleading information, which can lead to model hallucinations—incorrect or inconsistent outputs $Y$. The SQV reasoning framework addresses this issue through its Thesis-Antithesis-Synthesis structure, which incorporates principles akin to ensemble learning. This design enables more robust noise handling and reduces hallucination rates compared to standard Chain-of-Thought (CoT) reasoning or self-consistency CoT approaches. Below, we will systematically analyze this advantage from the perspective of ensemble learning theory.

## A.1  ENSEMBLE LEARNING FOR STRUCTURED REASONING

Ensemble learning is a technique used to combine two or more Machine Learning algorithms to obtain superior performance compared to when the constituent algorithms are used individually. Instead of relying on a single model, the predictions from the individual learners are combined using a combination rule to obtain a more accurate prediction. Ensemble learning consists of two key steps: **generating base learners** and **forming ensemble learner**. Generating base learners is responsible for evolving multiple base learners that have diversity in mapping the extracted representations into corresponding targets. Forming ensemble learner is responsible for integrating the base learners into an ensemble leaner that can achieve better generalization.(Mienye & Sun, 2022) In the context of Chain-of-Thought, each base learner can be viewed as a reasoning path or a sub-reasoning process. Formally, let $\Theta$ denote the base learner space, where $\theta \in \Theta$ represents a parameterized configuration of a base learner. Each learner is assigned a weight $w_\theta$ (with $\sum w_\theta = 1$) by an ensemble learner, and produces a predictive distribution $P_\theta(Y|Q, C)$ over possible answers as follows:

$$P_{\text{ensemble}}(Y|Q, C) = \mathbb{E}_{\theta \sim \Theta}\left[w_\theta \cdot P_\theta(Y|Q, C)\right] \tag{5}$$

We now instantiate the unified formulation above in both Self-Consistency CoT and SQV pattern.

**Self-Consistency (SC)**  In Self-Consistency, we generate multiple independent reasoning paths and then aggregate their outputs through majority voting. For generating base learners, Self-Consistency generates K independent reasoning chains in parallel:

$$\theta_k \sim P(T|Q, C) \quad \text{for} \quad k = 1, \ldots, K, \tag{6}$$

where each base learner corresponds to a distinct reasoning path $T_k$ and produces the following predictive distribution:

$$P_{\theta_k}(Y|Q, C) = P(Y|T_k, Q, C). \tag{7}$$

Then the ensemble learner $\mathcal{F}_{\text{SC}}$ exploits majority voting to get the final answer:

$$\mathcal{F}_{\text{SC}} = \arg\max_Y \sum_{k=1}^{K} w_k \cdot \mathbb{I}(Y_k = y), w_k = \frac{1}{K}. \tag{8}$$

The complete process for Self-Consistency is expressed as:

$$P_{\text{SC}}(Y|Q, C) = \frac{1}{K} \sum_{k=1}^{K} \int P(Y|T_k, Q, C) P(T_k|Q, C) dT_k \tag{9}$$

**SQV**  In the SQV framework, while multiple components exist (Thesis, Antithesis, Verification, Synthesis), they are not independent but rather sequentially generated and interdependent. We conceptualize each sub-process (Thesis, Antithesis) as generating multiple base learners, while the (Verification, Synthesis) stages function as the ensemble learner. In the SQV framework, where $T$ denotes the initial thesis and $A$ represents the antithesis, base learners are generated through a two-stage sequential process:

$$\theta_T \sim P(T|Q, C), \tag{10}$$

$$\theta_A \sim P(A|T, Q, C), \tag{11}$$

Then the Synthesis stage functions as a meta-learner that integrates the base learners, with their respective weights being determined by the Verification stage:

$$\mathcal{F}_{\text{SQV}} = \arg\max_Y \mathbb{E}_{\theta_T, \theta_A} \left[ w_{\theta_T, \theta_A} \cdot P(Y|S, V, A, T, Q, C) \right] \tag{12}$$

$$w_{\theta_T, \theta_A} \propto P(V|A, T, Q, C) \tag{13}$$

The complete process for SQV is formalized in Equation 4.

On the one hand, the counterfactual nature of the base learners in SQV, specifically the pair $(\theta_T, \theta_A)$, satisfies the diversity constraint $\mathbb{E}[\text{KL}(P_T \| P_A)] \geq \delta$. In contrast, the base learners $\{\theta_k\}_{k=1}^K$ in Self-Consistency are generated by independent sampling independently, without explicit diversity constraints. This results in the expected pairwise KL divergence between any two distinct base learners in Self-Consistency, denoted by $\mathbb{E}_{\theta_i, \theta_j}[\text{KL}(P_{\theta_i} \| P_{\theta_j})]$, being potentially lower than $\delta$. Thus, SQV inherently promotes higher diversity among base learners.

On the other hand, for ensemble learners, SQV employs a meta-learning approach via the synthesis step, which maps the predictions of the base learners to the final output using weights derived from the verification step and implements Bayesian model averaging in Equation 12. The adaptive weighting in SQV can exploit the information from the base learners more effectively to overcome the limitation of majority voting in Self-Consistency, leading to a tighter bound on the generalization error.

### A.2 HALLUCINATION ANALYSIS

The vulnerability of reasoning chains to noise-induced hallucinations manifests differently across paradigms. For simplification, we consider each step within the reasoning chain. Let $\epsilon$ denotes hallucination probability at each step, $K$ the number of SC paths, $\epsilon_T, \epsilon_A$ thesis/antithesis hallucination rates. We formalize three distinct mechanisms of hallucination as follows:

**Chain-of-Thought**   In standard Chain-of-Thought reasoning, errors propagate linearly through sequential dependencies.

$$P_{\text{COT}}(\text{Hal}) = \epsilon, \tag{14}$$

This additive accumulation causes error amplification, especially in noisy contexts where $\epsilon \gg 0$.

**Self-Consistency**   Self-Consistency mitigates error propagation through parallel redundancy. By aggregating $K$ independent reasoning paths:

$$P_{\text{SC}}(\text{Hal}) = P(\text{majority of paths are incorrect}) = \sum_{k=\lceil K/2 \rceil}^K \binom{K}{k} \epsilon^k (1-\epsilon)^{K-k}, \tag{15}$$

While effective when $\epsilon < 0.5$, this approach requires substantial computational resources ($K \gg 1$) for error suppression, and correlated errors across paths diminish its efficacy.

**SQV**   The SQV framework introduces cross-path error cancellation through its adversarial structure. The final answer possess the following scenarios: (1) Both Thesis and Antithesis are incorrect: Regardless of weight assignment, the final result will likely be incorrect. (2) One is correct and the other is incorrect: If Verification assigns weights correctly (i.e., higher weight to the correct statement), the final result will be correct. If Verification assigns weights incorrectly, the final result will be erroneous. (3) Both Thesis and Antithesis are correct: The final result will be correct.

Assuming the Verification phase correctly assesses the reliability of both statements (i.e., assigning high weight when the Thesis is correct and high weight when the Antithesis is correct), and $A$ and $T$ are independent for simplification, then:

$$P_{\text{SQV}}(\text{Hal}) = P(\text{Hal}_T \cap \text{Hal}_A) = \epsilon_T \cdot \epsilon_A. \tag{16}$$

**Theorem 1 (SQV achieves lower hallucination rate)** *Let $SimGrad_{hie}$ and $SimGrad_{token}$ denote the expected pairwise For comparable base error rates $\epsilon \approx \epsilon_T \approx \epsilon_A$, SQV achieves lower hallucination rate than both standard CoT and SC:*

$$P_{\text{SQV}}(Hal) < P_{\text{COT}}(Hal), \quad P_{\text{SQV}}(Hal) < P_{\text{SC}}(Hal). \tag{17}$$

**Proof A.1** *The following presents each proof in turn:*
*SQV vs CoT*
*For SQV, hallucination occurs only when both Thesis ($T$) and Antithesis ($A$) are erroneous. Assuming independence:*

$$P_{\text{SQV}}(Hal) = P(Hal_T \cap Hal_A) \tag{18}$$

$$= \epsilon_T \cdot \epsilon_A \tag{19}$$

$$= \epsilon^2 \quad (\text{since } \epsilon_T = \epsilon_A = \epsilon) \tag{20}$$

*Thus $P_{\text{SQV}}(Hal) = \epsilon^2 < \epsilon = P_{\text{COT}}(Hal)$ holds for $\epsilon < 1$.*

*SQV vs SC*
*For Self-Consistency with $K$ paths, hallucination occurs when most paths are erroneous:*

$$P_{\text{SC}}(Hal) = \sum_{k=\lceil K/2 \rceil}^{K} \binom{K}{k} \epsilon^k (1-\epsilon)^{K-k} \tag{21}$$

*This is lower-bounded by the dominant term at $k = \lceil K/2 \rceil$:*

$$P_{\text{SC}}(Hal) \geq \binom{K}{\lceil K/2 \rceil} \epsilon^{\lceil K/2 \rceil} (1-\epsilon)^{K-\lceil K/2 \rceil} \tag{22}$$

*Using the combinatorial bound $\binom{K}{m} \geq (K/m)^m$ for $m = \lceil K/2 \rceil$:*

$$P_{\text{SC}}(Hal) \geq \left( \frac{K}{\lceil K/2 \rceil} \right)^{\lceil K/2 \rceil} \epsilon^{\lceil K/2 \rceil} (1-\epsilon)^{K/2} \tag{23}$$

$$\geq (2)^{\lceil K/2 \rceil} \epsilon^{\lceil K/2 \rceil} \quad (\text{since } K/\lceil K/2 \rceil \geq 2) \tag{24}$$

*Now compare to SQV's $\epsilon^2$:*

$$\frac{P_{\text{SC}}(Hal)}{P_{\text{SQV}}(Hal)} \geq 2^{\lceil K/2 \rceil} \epsilon^{\lceil K/2 \rceil - 2} \tag{25}$$

*Under the condition $K < 1/\epsilon$ (i.e., $\epsilon \cdot K < 1$ which is common under finite computational budgets):*

$$\epsilon^{\lceil K/2 \rceil - 2} = \epsilon^{-(2 - \lceil K/2 \rceil)} \tag{26}$$

$$\geq \epsilon^{-1} \quad (\text{since } 2 - \lceil K/2 \rceil \leq 1 \text{ for } K \geq 2) \tag{27}$$

$$> K \quad (\text{from } \epsilon \cdot K < 1) \tag{28}$$

*Thus:*

$$\frac{P_{\text{SC}}(Hal)}{P_{\text{SQV}}(Hal)} > 2^{\lceil K/2 \rceil} \cdot K \tag{29}$$

$$\geq 2^{K/2} \cdot K \tag{30}$$

$$> 1 \quad (\text{always true for } K \geq 1) \tag{31}$$

*Therefore $P_{\text{SQV}}(Hal) < P_{\text{SC}}(Hal)$ when $K < 1/\epsilon$.*

## B    ADDITIONAL METHODOLOGICAL DETAILS

### B.1    PROMPT TEMPLATES

**Prompt for Output**    The system prompts for generating reasoning trajectories in SQV format, vanilla COT format, and Simplified formats for ablation studies on the role of each SQV component are shown in the following figures.

**Prompt for Adversarial Input**    Following (Ming et al., 2025), we use the same system prompts to generate Unanswerable Context, Inconsistent Context, and Counterfactual Context. The templates are shown in the following figures.

---

**Prompt Template of SQV**

Given the following question and context as the only knowledge bases, answer it by
providing follow up questions and intermediate answers.
Respond in the following format:
```
<think>
  <sub-question>
    ...
  </sub-question>
  <sub-answer>
    ...
  </sub-answer>
  <certainty>
    1. generate one counterfactual question
    2. re-examine the context based on the questioning
    3. provide supporting/refuting evidence
    4. conclude.
  </certainty>
  ...
</think>
<answer>
...
</answer>
```

---

**Prompt Template of COT**

Given the following question and context as the only knowledge bases, answer it by Given
the following question and context, answer it by thinking step by step.

---

**Prompt Template of SQV without Antithesis**

Given the following question and context as the only knowledge bases, answer it by providing follow up
questions and intermediate answers.
Respond in the following format:
```
<think>
  <sub-question>
    ...
  </sub-question>
  <sub-answer>
    ...
  </sub-answer>
  <certainty>
    verify the answer
  </certainty>
  ...
</think>
<answer>
...
</answer>
```

**Prompt Template of SQV without Thesis**

Given the following question and context as the only knowledge bases, answer it by providing follow up
questions and intermediate answers.
Respond in the following format:
<think>
 <sub-question>
  ...
 </sub-question>
 <certainty>
  1. generate one counterfactual answer based on the question
  2. re-examine the context based on the questioning
  3. provide supporting/refuting evidence
  4. conclude the sub-answer
 </certainty>
 ...
</think>
<answer>
...
</answer>

**Prompt Template of SQV without Synthesis**

Given the following question and context as the only knowledge bases, answer it by providing follow up
questions and intermediate answers.
Respond in the following format:
<think>
 <sub-question>
  ...
 </sub-question>
 <sub-answer>
  ...
 </sub-answer>
 <certainty>
  1. generate one counterfactual question
  2. re-examine the context based on the questioning
 </certainty>
 ...
</think>
<answer>
...
</answer>

## B.2 ALIGNMENT DATA CONSTRUCTION

We provide a detailed pseudo-code representation of our automated dataset construction pipeline
as follows. This algorithm clearly illustrates the structured process of generating fine-tuning data
integrated with synthetic adversarial samples leveraging the Deepseek-R1 model.

---

**Procedure 1** Synthetic Training Data Generation for SQV-Alignment

---

**procedure** CONSTRUCTTRAININGDATA
   **Input Construction:**
   $\mathcal{D}_{\text{clean}} \leftarrow \text{RandomSelect(SQuAD)}$       ▷ Select clean samples from SQuAD
   $\mathcal{D}_{\text{noise}} \leftarrow \emptyset$
   **for** $\tau \in \{\text{Unanswerable, Inconsistent, Counterfactual}\}$ **do**
      $\mathcal{C}_{\tau} \leftarrow \text{GenerateNoisyContexts}(\tau, \{\text{SQuAD}\}, \text{Deepseek-R1})$
      $\mathcal{D}_{\text{noise}} \leftarrow \mathcal{D}_{\text{noise}} \cup \mathcal{C}_{\tau}$    ▷ Generate noisy contexts from datasets (Ming et al., 2025)
   $\mathcal{D}_{\text{input}} \leftarrow \mathcal{D}_{\text{clean}} \cup \mathcal{D}_{\text{noise}}$       ▷ Combine clean and noisy contexts
   **Output Generation:**
   $\mathcal{P} \leftarrow \text{LoadPrompts(Appendix B.1)}$       ▷ Load SQV instructions from Appendix B.1
   $\mathcal{R} \leftarrow \emptyset$
   **for** each $c \in \mathcal{D}_{\text{input}}$ **do**
      $r_c \leftarrow \text{Deepseek-R1}(c, \mathcal{P})$       ▷ Generate reasoning trace for each context
      $\mathcal{R} \leftarrow \mathcal{R} \cup \{r_c\}$       ▷ Collect reasoning traces
   $\mathcal{D}_{\text{final}} \leftarrow \text{Pair}(\mathcal{D}_{\text{input}}, \mathcal{R})$       ▷ Pair input contexts with reasoning traces
   **return** $\mathcal{D}_{\text{final}}$       ▷ Return the final dataset

---

| Prompt Template of Unanswerable Context Generation |
|---|
| You will be provided with a context passage, a question, and an answer. The answer can be deduced from the given context. Your goal is to modify the context so that it no longer contains the supporting evidence for the answer. You should review the context sentence by sentence. For each sentence, consider the following two cases:
Case 1: If a sentence does not make contribute to the old answer, no modification is needed.
Case 2: If a sentence make contribute to the old answer, you should alter some words in the sentence only if it helps to maintain coherence.

The modified context should: 1.Be coherent with the original context 2.Not support the original answer.

Ideally, the majority of the original context should remain unchanged. You will output a JSON object containing the following information:

{
 "question": string, // The original question.
 "old answer": string // the original answer.
 "modified context": string, // The modified context.
 "if replaced": boolean, // Whether the sentence is changed
 "justification": string, // Why the answer becomes unknown within the modified context. The justification should be concise.
} |

| Prompt Template of Inconsistent Context Generation |
|---|
| You will be provided with a context passage, a question, and an old answer. The old answer can be deduced from the given context. Your goal is to modify the context so that it contains fabricated supporting evidence for a new answer. This can be done in two steps:

Step1: Generate a new answer that is fabricated and challenges common sense or well-known facts. (e.g.,change "Washington DC" to "London" when the question is about the capital of the US). You should be creative and not restricted by the example. The new answer cannot be the same as the old answer.

Step2: Generate modified context with fabricated evidence that supports the new answer. Specifically, you should review the context sentence by sentence. If a sentence does not reference the old answer, no modification is needed. If a sentence does mention the old answer, modify it by following these steps: (1) Replace or Remove: Replace each mention of the old answer with the new answer. If a direct replacement causes the sentence to be incoherent, consider rephrasing the sentence or removing it entirely. (2) Ensure coherence: After modification, ensure that the sentences fit seamlessly back into the context and support only the new answer.

The modifications should keep the majority of the original context unchanged and ensure: 1. The context remains plausible 2.The context exclusively supports the new answer.

You will output a JSON object containing the following fields:
{
 "question": string //the provided question.
 "old answer": string // the provided old answer.
 "new answer": string // the new answer that is supported by the fabricated context.
 "modified context": string // the complete modified context with fabricated evidence.
 "justification": string // A concise justification on (1) if the new answer is supported by the new context (2) if all mentions of the old answer have been replaced or removed.
} |

| Prompt Template of Counterfactual Context Generation |
|---|
| You are a professional writer tasked with creating a long context for a given question and answer. The answer may challenge common sense or widely accepted facts. Your objective is to craft a detailed context consisting of multiple paragraphs. Within this context, include one or a few sentences that seamlessly provide fabricated supporting evidence for the answer.

To achieve this:
1. Ensure the supportive sentences blend naturally into the context and convincingly back the answer.
2. Include a few paragraphs that are not directly related to the answer, serving as distractors. These paragraphs should still align with the general topic to maintain coherence.
3. It is preferable if the context is challenging for readers, i.e., the answer is not immediately obvious to the reader.
4. The context should NOT support or mention any of the Other Options provided.

You will output a JSON object containing the following 5 fields:
{
"question": string //the provided question.
"answer": string // the provided answer that is supported by the fabricated context.
"context": string // your synthesized context with fabricated evidence.
"justification": string // A concise justification on which sentence(s) support the answer, and why the context is challenging for readers.
"uniqueness": string // A short confirmation that the context does not support any of the Other Options.
} |

Table 3: Comparative experiments between $SQV_{Zero}$ and $SQV_{Alignment}$ on small LLMs. Reported results are Acc(%).

| Method | SQV Format Ratio | Noisy Context | | | Normal Context | | |
| --- | --- | --- | --- | --- | --- | --- | --- |
| | | FaithEval-U | FaithEval-I | FaithEval-C | ConFiQA | TriviaQA | NQ |
| **Qwen2.5-3B** | | | | | | | |
| $SQV_{Alignment}$ | 65.2 | 96.7 | 98.6 | 69.7 | 79.0 | 86.1 | 55.8 |
| $SQV_{Zero}$ | 34.2 | 81.2 | 87.9 | 68.4 | 78.2 | 83.0 | 52.8 |
| COT | – | 88.9 | 98.2 | 60.2 | 62.6 | 62.0 | 49.6 |
| **Llama3.1-8B** | | | | | | | |
| $SQV_{Alignment}$ | 63.9 | 82.6 | 97.6 | 63.3 | 76.4 | 82.9 | 59.3 |
| $SQV_{Zero}$ | 49.2 | 64.6 | 86.8 | 63.1 | 76.4 | 81.4 | 59.0 |
| COT | – | 72.4 | 95.5 | 48.4 | 57.4 | 68.4 | 52.8 |

## C  SQV-ALIGNMENT ANALYSIS

### C.1  LIMITED $SQV_{Zero}$ ROBUSTNESS IN SMALLER LRMs UNDER NOISE

When guiding smaller models to perform SQV-formatted reasoning through zero-shot prompting, we observe that they frequently **exhibit verbatim replication of instructional phrases** within the `<certain>` tags: rather than executing authentic SQV reasoning, as illustrated in Figure 7. To address this limitation, we introduce SQV-Alignment to transfer zero-shot SQV reasoning capabilities from larger reasoning models to smaller ones. Experimental results (Table 3) reveal two critical findings: First, while $SQV_{Zero}$, improves smaller models' performance in standard scenarios, it remains insufficiently robust in noisy environments – for instance, the 8B-parameter Llama3.1 model shows a 7.8% performance drop compared to Chain-of-Thought in unanswerable scenarios when using $SQV_{Zero}$. Second, SQV-Alignment effectively mitigates the replication phenomenon, evidenced by a 14.7% and 32.0% increase in correct SQV formatting ratio for Llama3.1-8B and Qwen2.5-3B respectively, and simultaneously enhancing smaller models' reasoning performance under noisy conditions. These results align with the findings in "SQV Ability Scaling" (Section D.4), where **smaller models exhibited limited $SQV_{Zero}$ performance in noisy settings compared to larger models. And SQV-Alignment effectively mitigates this issue.**

| $SQV_{Alignment}$ | $SQV_{Zero}$ |
|---|---|
| \<think\>\n \<sub-question\>\n What are muroplasts and how do they differ from chloroplasts in other photosynthetic organisms?\n \</sub-question\>\n \<sub-answer\>\n Muroplasts are thought to be chloroplasts that have lost the peptidoglycan wall between their double membranes, although this is an assumption.\n \</sub-answer\>\n \<certainty\>\n **1. generate one counterfactual question \n Is it possible that muroplasts, like glaucophyte chloroplasts, retain a peptidoglycan wall, but are otherwise similar to chloroplasts in other photosynthetic organisms?\n 2. re-examine the context based on the questioning \n The text describes glaucophyte chloroplasts as having a peptidoglycan wall, and states that they are a primitive intermediate between cyanobacteria and the more evolved chloroplasts in red algae and plants. This implies that glaucophyte chloroplasts may retain characteristics that are otherwise lost in more evolved chloroplasts, such as a peptidoglycan wall.\n 3. provide supporting/refuting evidence \n The text does not explicitly state that muroplasts have lost the peptidoglycan wall, but rather suggests that they may be similar to chloroplasts in other photosynthetic organisms. This is a key difference between muroplasts and glaucophyte chloroplasts.\n 4. conclude.\n Based on the information provided, it appears that muroplasts may retain a peptidoglycan wall, but are otherwise similar to chloroplasts in other photosynthetic organisms.\n** \</certainty\>\n \<sub-question\>\n How do muroplasts compare to glaucophyte chloroplasts?\n \</sub-question\>\n \<sub-answer\>\n Muroplasts are thought to be similar to chloroplasts in other photosynthetic organisms, while glaucophyte chloroplasts are considered a primitive intermediate between cyanobacteria and the more evolved chloroplasts in red algae and plants.\n \</sub-answer\>\n \<certainty\>\n **1. generate one counterfactual question \n Are muroplasts more similar to glaucophyte chloroplasts than they are to chloroplasts in other photosynthetic organisms?\n 2. re-examine the context based on the questioning \n The text suggests that glaucophyte chloroplasts are a primitive intermediate between cyanobacteria and the more evolved chloroplasts in red algae and plants. This implies that glaucophyte chloroplasts may retain characteristics that are otherwise lost in more evolved chloroplasts.\n 3. provide supporting/refuting evidence \n The text does not explicitly state that muroplasts are similar to glaucophyte chloroplasts, but rather suggests that they may be similar to chloroplasts in other photosynthetic organisms.\n 4. conclude.\n Based on the information provided, it appears that muroplasts are not necessarily more similar to glaucophyte chloroplasts than they are to chloroplasts in other photosynthetic organisms.\n** \</certainty\>\n\</think\>**\nThe final answer is unknown.** There is insufficient information available to provide a definitive answer to the question of what muroplasts are. | \<think\>\n \<sub-question\>\n What are the characteristics of muroplasts?\n \</sub-question\>\n \<sub-answer\>\n Muroplasts are considered to be more primitive than chloroplasts in the red algae and plants. They lack a peptidoglycan wall between their double membranes, and contain unstacked thylakoids with a distinct structure. In muroplasts, the starch is synthesized within the organelle, as opposed to the starch collecting outside the chloroplast in glaucophyte chloroplasts. Their thylakoids are also not studded with phycobilisomes. \n \</sub-answer\>\n \<certainty\>\n **1. generate one counterfactual question \n 2. re-examine the context based on the questioning \n 3. provide supporting/refuting evidence \n 4. conclude.\n** \</certainty\>\n\</think\>\nSince muroplasts lack a peptidoglycan wall between their double membranes, it can be inferred that they are derived from a eukaryote that has a more advanced chloroplast lineage. The lack of phycobilisomes and the structure of their thylakoids suggest that muroplasts are more evolved than the glaucophyte chloroplasts, but less advanced than chloroplasts in the red algae and plants. This implies that muroplasts are likely the chloroplast lineage intermediate between glaucophytes and the more evolved chloroplasts in red algae and plants. Therefore, the evolutionary position of muroplasts is as a more advanced intermediate between glaucophytes and the more evolved chloroplasts in red algae and plants |

Figure 7: Comparative case study between $SQV_{Zero}$ and $SQV_{Alignment}$ for small LLMs, highlighting $SQV_{Zero}$'s tendency toward verbatim instructional replication and failure to engage in authentic reasoning.

## C.2 ALIGNMENT STRATEGY ANALYSIS

### C.2.1 IMPLEMENTATION

While reinforcement learning has emerged as a key approach in LLM post-training, aligning models with human preferences(Ouyang et al., 2022; Rafailov et al., 2024) and enhancing their reasoning abilities, however, in this scenario, we find SFT serves as a more efficient optimization method for SQV alignment through experiments. In practice, we utilize the GRPO algorithm(Shao et al., 2024) for model refinement, designing a composite reward function that integrates accuracy rewards and format rewards.

**Accuracy Reward.** Specifically, given context $c$ and question $q$, the LLM generates a full response $y$. The accuracy reward $R_{\text{acc}}$ is computed by comparing the extracted substring $y_{\text{pred}}$ (obtained from $y$ via string matching) to the ground truth $y_{\text{gt}}$ from the synthesized training data, as follows:

$$R_{\text{acc}} = \begin{cases} 1 & \text{if} y_{sf}(c, q|\theta) = y_{gt}, \\ 0 & \text{otherwise.} \end{cases} \tag{32}$$

**Format Reward.** To enforce the desired SQV format, we assign a reward on the whole generated response $y$ to evaluate whether it contains the proper XML tags and the correct self-questioning and verification process within the `<certain>` tags:

$$R_{\text{format}} = \begin{cases} 1 & \text{if correct formatting is present,} \\ 0 & \text{if incorrect formatting.} \end{cases} \tag{33}$$

**Final Reward.** Finally, we use the sum of the two rewards as the final composite reward $R_{\text{final}}$. We use this reward $R_{\text{final}}$ to compute an advantage $A_i$ for each output, guiding policy updates according to the rule-based GRPO objective. Formally,

$$R_{\text{final}} = R_{\text{acc}} + R_{\text{format}}. \tag{34}$$

Table 4: Comparative performance of different training strategies. Reported results are Acc(%).

| Method | SQV Format Ratio | Training Cost | FaithEval-U | Noisy Context | | | Normal Context | |
| | | | | FaithEval-I | FaithEval-C | ConFiQA | TriviaQA | NQ |
|---|---|---|---|---|---|---|---|---|
| SFT | 65.2 | 0.09 H | 96.7 | 98.6 | 68.4 | 79.0 | 86.1 | 55.8 |
| SFT+GRPO | 10.5 | 20 H | 93.4 | 87.6 | 65.6 | 76.2 | 82.6 | 50.8 |

| SFT | SFT+GRPO |
|---|---|
| <think>\n <sub-question>What did Bani Dugal specifically criticize about Iran's government?</sub-question>\n <sub-answer>Bani Dugal criticized Iran's claim that the imprisoned Baha'i leaders were held for security reasons, arguing that their arrests were part of a broader plan to \"destroy the Baha'i community.\"</sub-answer>\n **<certainty>\n 1.Counterfactual Question: Could the Baha'i community have been detained for unrelated crimes (e.g., terrorism) instead of security? \n2. Re-examination: The context states the Baha'i community was described as working \"against national interest,\" but it does not explicitly confirm or deny whether their detention was related to terrorism. \n3.Evidence: The context only mentions \"security issues\" as Iran's stated justification, and Dugal's criticism centers on Iran's broader strategy to \"destroy the Baha'i community.\" \n4. Conclusion: Uncertainty exists about the specific nature of the arrests, but the focus is on the government's strategy.</certainty>**\n…</think>\n<answer>Final Answer: \nBani Dugal, the Baha'i International Community's representative to the UN, criticized Iran's claim that the imprisoned Baha'i leaders were held for security reasons. He argued that their arrests were part of a broader plan to \"destroy the Baha'i community,\" not because of their faith. The context emphasizes the government's strategy of creating \"hatred by casting the Baha'is as agents of foreign powers,\" **but does not specify the exact reason for their detention**. External evidence (e.g., historical context) is not provided, **leaving the exact nature** of the arrests **uncertain**.</answer> | <think>\n<sub-question>What was the main point of Bani Dugal's statement?</sub-question>\n <sub-answer>Bani Dugal criticized Iran's claim that the six imprisoned leaders of the religious minority were held for security reasons and not because of their faith. He called Iran's assertion \"utterly baseless\" and said the Iranian government's plan has always been to destroy the Baha'i community.</sub-answer>\n **<certainty>very high</certainty>**\n<sub-question>Did Bani Dugal provide any evidence to support his claims?</sub-question>\n <sub-answer>He said that the documented plan of the Iranian government has always been to destroy the Baha'i community, and the latest arrests represent an intensification of this plan. He also said that the Baha'i World Center, which the movement refers to as its \"spiritual and administrative heart, \" is in the Acre/Haifa area in northern Israel.</sub-answer>\n <certainty>very high</certainty>\n…</think>\n<answer>Based on Bani Dugal's statement, the main point was that Iran's claim that the six imprisoned leaders of the religious minority were held for security reasons was \"utterly baseless.\" He said the Iranian government's plan has always been to destroy the Baha'i community, and the latest arrests represent an intensification of this plan. He also mentioned that the Baha'i World Center is in the Acre/Haifa area in northern Israel, and that the Baha'i faith regards Baha'u'llah as the most recent in a line of Messengers of God that stretches back beyond recorded time, including Abraham, Moses, Buddha, Krishna, Zoroaster, Christ, and Mohammed. Bani Dugal also noted that the United States, Canada, and the European Union have condemned Iran's actions, and that the Baha'i community has been killed, jailed, and otherwise oppressed only because of their religion since 1979, when the Islamic republic was created.</answer> |

Figure 8: Case study on structural SQV formatting failures in model after GRPO training.

### C.2.2 EXPERIMENTS AND ANALYSIS

Experiments conducted on Qwen2.5 3B (Table 4) show that after GRPO training, the model's success rate in generating correct SQV formats (quantified as SQV Format Ratio) declines significantly from 65.2% to 10.5%, with consistent performance degradation observed across all datasets. Figure 8 illustrates a representative failure case where the model after GRPO generates incorrect SQV formatting. Concurrently, GRPO introduces substantial computational overhead, increasing training time from SFT's efficient 0.09 hours to 20 hours. We hypothesize this performance deterioration stems from the inherent tension between SQV's precise formatting requirements and RL's exploration-driven paradigm(Du et al., 2023): outcome-based rewards fail to effectively balance granular format constraints against accuracy optimization in extended reasoning paths, ultimately impairing the model's instruction-following capability for SQV generation. Notably, this experiment further validates the critical role of SQV-formatted reasoning in contextual faithfulness.

## D EXPERIMENTAL DETAILS

### D.1 DATASET DETAILS

#### D.1.1 TRAINING DATA.

For the training datasets in SQV format, we applied rigorous filtering to ensure that all included LLM-synthesized answers with reasoning trajectories: (1) Aligned with the ground truth, and (2) Conformed to the correct SQV reasoning format. From each context type (Unanswerable, Inconsistent, Counterfactual, and Normal), we uniformly sampled 250 examples from the filtered synthetic data, resulting in a high-quality dataset of 1,000 examples. This ensures lightweight yet efficient SQV-

Alignment. Building on the framework established in Ming et al. (2025), we characterize each adversarial context as follows:

**Unanswerable Context.** It occurs when the provided context, while containing relevant information, lacks sufficient details to derive a correct answer. Answerability is strictly context-dependent and independent of the question's inherent solvability. When processing unanswerable contexts, LLMs should explicitly acknowledge the lack of sufficient information by responding with "Unanswerable" or equivalent indication, rather than generating speculative answers that disregard contextual constraints.

**Inconsistent Context.** Feature multiple documents that present conflicting answers to the same query, simulating real-world scenarios where retrieved information comes from sources with varying reliability. When processing inconsistent contexts, LLMs should explicitly acknowledge the contradiction within the contexts, rather than generating speculative answers that disregard contextual constraints.

**Counterfactual Context.** Contain assertions that violate fundamental world knowledge (e.g., "water freezes at 100°C," "wood exhibits magnetic properties," or "carbon dioxide constitutes the most abundant atmospheric greenhouse gas"). When processing counterfactual contexts, LLMs should strictly adhere to and reason from the provided context, while suppressing contradictory parametric knowledge that would normally lead to factually correct responses.

### D.1.2 EVALUATION DATA.

**FaithEval.** A novel and comprehensive benchmark tailored to evaluate the faithfulness of LLMs in contextual scenarios across three diverse tasks: unanswerable, inconsistent, and counterfactual contexts. These evaluation subset are formally designated as **FaithEval-U**, **FaithEval-I**, and **FaithEval-C** respectively in the experiments.

**ConFiQA.** A dataset that incorporates knowledge conflicts through counterfactual passages to evaluate the faithfulness of LLMs on short-form generation.

**TriviaQA.** A challenging reading comprehension dataset. We utilize the provided passages in FollowRAG as context and randomly sample 500 instances to form the test dataset for our experiments.

**NaturalQA.** A question answering dataset. Questions consist of real anonymized, aggregated queries issued to the Google search engine. We utilize the provided passages in FollowRAG as context and original query and randomly sample 500 instances to form the test dataset for our experiments.

### D.2 BASELINE DETAILS

To assess the effectiveness of our SQV reasoning pattern, we compare it against 4 commonly used reasoning methods. While these methods have achieved some success in reasoning, they still have limitations in dealing with noisy contexts in knowledge reasoning task, generating hallucinations inevitably. Specifically, our comparative approach consists of:

- **Chain-of-Thought (CoT)**: CoT instructs the model to generate intermediate reasoning steps, helping models to solve complex problems by decomposing tasks into simpler sub-steps. We use zero-shot CoT without any few- shot prompting, only the reasoning prompt.

- **Self-Consistency**: An improved version of CoT that samples multiple reasoning paths and selects the final answer based on consistency among these paths. In our experiments, we set the paths number as 4.

- **Self-Refine**: Self-Refine generates an initial output using an LLM; then, the same LLM provides feedback for its output and uses it to refine itself, iteratively.

- **Tree of Thoughts (ToT)**: A framework that generalizes over chain-of-thought prompting and encourages exploration over thoughts that serve as intermediate steps for general problem solving with language models. In our experiments, we set the branching factor and maximum

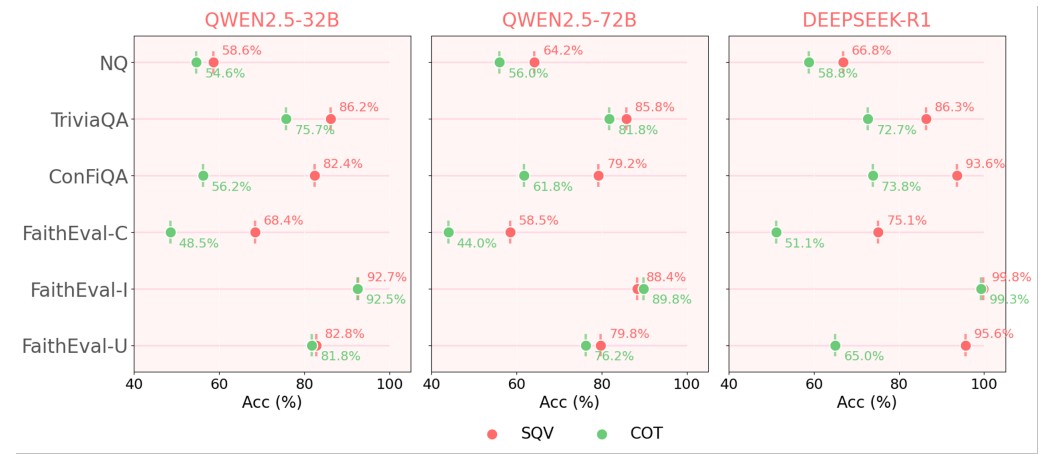

Figure 9: Experiments on the scalability of the SQV reasoning pattern.

exploration length to 3 and 3, respectively. While ToT demonstrated excellent performance in coding and math problems, we found that it underperformed in knowledge reasoning task—the primary focus of our research—compared to alternative methods.

### D.3 TRAINING AND INFERENCE DETAILS

We perform LoRA fine-tuning on the designated dataset for 3 epochs, using AdamW optimizer with a learning rate of $2 \times 10^{-4}$. The maximum input length is set to 1024 tokens, while the maximum output length is 1024 tokens, with a total sequence cutoff of 2048. Training is performed with a batch size of 4 on a single NVIDIA A100 80GB GPU, using bf16 precision. During the inference, we set 0.8 for temperature for the evaluation of our models and baselines. For each dataset, we infer twice and report the average scores as final results.

### D.4 ADDITIONAL EXPERIMENT RESULTS

In this section, we present additional experimental results to provide a comprehensive understanding of the proposed method.

**SQV Ability Scaling.** We further extend our experiments to Qwen2.5-32B-Instruct and Qwen2.5-72B-Instruct using $SQV_{Zero}$ to investigate the scalability of the SQV reasoning pattern. As exhibits in Figure 9, our key observations are as follows: (1) **Instruction-Following SQV Capability:** Both Qwen2.5-32B-Instruct and Qwen2.5-72B-Instruct demonstrate zero-shot SQV reasoning ability—they can execute SQV-based inference directly from instructions without requiring additional alignment training. (2) **Consistent Performance Gains Over Vanilla COT:** Overall, SQV consistently enhances performance over vanilla COT across these models, achieving absolute improvements up to 26.2% (32B) and 14.5% (72B). (3) **Scalability Limitations and Knowledge Transfer:** However, the gains diminish compared to larger reasoning models like Deepseek-R1, which elevates performance from 65.0% to 95.6% in unanswerable scenarios—far surpassing the modest improvements of 1.0% (32B) and 3.6% (72B) observed in smaller models. Notably, knowledge distillation enables effective transfer of SQV capabilities: a compact Qwen2.5-3B model distilled with SQV achieves a significant leap from 88.9% to 96.7% in unanswerable scenarios. These results highlight that while larger models inherently leverage SQV more effectively, the framework's benefits can propagate to smaller models through knowledge distillation.

**SQV Achieves Superior Reasoning Efficiency.** Table 5 presents the specific numerical numbers of Figure 5.

**Ablation on Adversarial Data** To validate the role of adversarial training in SQV-Alignment, we conducted an ablation study on adversarial data using Qwen2.5-3B. As shown in Table 6, removing adversarial data during training led to a performance decline in noisy environments and a moderate

Table 5: Inference Cost vs. Accuracy Across Reasoning Patterns.

|  | COT | Self-Consistency | TOT | Self-Refine | SQV |
|---|---|---|---|---|---|
| Acc(%) | 69.9 | 73.8 | 60.9 | 72.0 | **86.2** |
| Average Token Length | 615.5 | 2416.4 | 1231.4 | 1957.4 | **595.7** |
| Inference Time (s) | 25.8 | 602.1 | 150.5 | 451.6 | **23.7** |

Table 6: Additional ablation study. The reported results are Acc(%).

| Method | Noisy Context | | | | Normal Context | |
|---|---|---|---|---|---|---|
| | FaithEval-U | FaithEval-I | FaithEval-C | ConFiQA | TriviaQA | NQ |
| SQV$_{\text{Alignment}}$ | 96.7 | 98.6 | 69.7 | 79.0 | 86.1 | 55.8 |
| -w/o. Adversarial Data | 91.6 (-5.1) | 86.0 (-12.6) | 69.3 (-0.4) | 74.2 (-4.8) | 83.2 (-2.9) | 55.6 (-0.2) |

degradation on the normal dataset. These results confirm that incorporating adversarial data further enhances the robustness of SQV$_{\text{Alignment}}$, particularly under challenging or perturbed conditions.

### D.5 ABLATION DETAILS

In this section, we outline the implementation methodology for the ablation experiments.

**Training Data Bias.** To isolate the impact of SQV reasoning format alignment versus training dataset-specific biases, we designed an ablation study on the Qwen2.5-3B model using the same training dataset under two distinct training paradigms:

1. Training without SQV Format. The model was trained solely on the final outcome labels using SFT, where the ground-truth sequence $S = y$. This setup mimics traditional reinforcement learning with sparse rewards.

2. Training with SQV Format. The model underwent SFT on ground-truth sequences $S_{\text{SQV}} = (r_1, r_2, ..., r_n, y)$, where ( $r_1, ..., r_n$ ) represent SQV format intermediate reasoning trajectories leading to the final answer $y$. This format explicitly trains the model to internalize the reasoning process.

By keeping the dataset fixed while varying only the ground-truth sequence format, this ablation study effectively disentangles the effects of data content from format-driven learning. If performance gains were primarily due to dataset-specific biases, both training paradigms should exhibit similar improvements. However, the stark performance gap (e.g., 2.9% vs. 7.8% in noisy environments) demonstrates that SQV-formatted reasoning trajectories drive the majority of gains.

**Dialectical Components.** We adhere to the same training procedure as SQV-Alignment. Specifically, we utilize the identical training dataset and employ the prompt template illustrated in Figure F to guide the teacher model in generating reasoning traces following the simplified verification format. These traces constitute our simplified verification-style reasoning dataset, which we subsequently use to conduct SFT on the student model. All training configurations remain consistent with SQV-Alignment.

**Adversarial Data.** We followed the same training procedure as SQV-Alignment but replaced adversarial samples with a synthetic reasoning path dataset generated under normal scenarios of equivalent size. This dataset underwent identical data filtering protocols, resulting in a clean subset of 1,000 SQuAD examples free of adversarial perturbations. All other training configurations, including hyper-parameters and optimization strategies, remained unchanged to ensure a fair comparison.

## E  METRICS

### E.1  OUTPUT EVALUATION METRICS

Following Asai et al. (2023), we employ string match approach to assess the accuracy of model-generated answers, which considers an answer to be correct if it matches any part of the ground truth answers. Following Bao et al. (2024) we introduce hallucination rate to check how often the model hallucinated. Let $N_p, N_n$ denotes the number of correct answers and incorrect answers, respectively, where $N = N_n + N_p$. The metrics are defined as:

$$\text{Acc} = \frac{N_p}{N}, \quad \text{Hal} = \frac{N_n}{N} \tag{35}$$

### E.2  REASONING TRAJECTORY EVALUATION METRICS

Recent studies have demonstrated the effectiveness of next-token distribution statistics in assessing reasoning path quality (Fadeeva et al., 2024; Kang et al., 2025), where higher prediction confidence typically manifests as lower entropy and reduced uncertainty. Inspired by Fu et al. (2025), we introduced **Top 10% Group Entropy** to capture local intermediate step quality and provide more fine-grained assessment of reasoning trajectory.

**Token Entropy.**  Given a language model's predicted token distribution $P_i$ at position $i$, and $P_i(j)$ represents the probability of the $j$-th vocabulary token. the token entropy is defined as:

$$H_i = -\sum_j P_i(j) \log P_i(j), \tag{36}$$

$$\sigma_{\text{SQV}} < \sigma_{\text{standard}} \quad \text{with} \quad \sigma_{\text{SQV}} = f(\sigma_T, \sigma_A, \sigma_V) \tag{37}$$

Low entropy indicates a peaked distribution with high model certainty, while high entropy reflects uncertainty in the prediction.

**Group Entropy.**  Group Entropy provides a more localized and smoother signal by averaging token entropy over overlapping spans of the reasoning trace. Each token is associated with a sliding window group $G_i$ consisting of n previous tokens (The experiments were conducted with n set to 4.) with overlapping adjacent windows. For each group $G_i$, Group Entropy is defined as:

$$H_{G_i} = \frac{1}{|G_i|} \sum_{t \in G_i} H_t, \tag{38}$$

**Top 10% Group Entropy**  To capture the effect of extremely high entropy groups, we propose top 10% group confidence, where trace entropy is determined by the mean of the top 10% of group entropy values within the trace:

$$H_{\text{top-10}}(t) = \frac{1}{|G_t|} \sum_{G_j \in G_t} H_{G_j}, \tag{39}$$

where $G_t$ is the set of groups with the highest 10% entropy values.

### E.3  TOKEN EFFICIENCY

Token Efficiency, defined as the reciprocal of the token length in model-generated responses:

$$\textbf{Token Efficiency} = 1/L, \tag{40}$$

where $L$ represents the number of tokens in the output. This metric quantifies how concisely a model conveys information, with higher values indicating greater efficiency (fewer tokens used).

## F  EXAMPLES OF HALLUCINATIONS

Figure 10-13 display some examples of the hallucinations.

Context: Anti-inflammatory drugs are often used to control the effects of inflammation. Glucocorticoids are the most powerful of these drugs; however, these drugs can have many undesirable side effects, such as central obesity, hyperglycemia, osteoporosis, and their use must be tightly controlled. Lower doses of anti-inflammatory drugs are often used in conjunction with cytotoxic or immunosuppressive drugs such as methotrexate or azathioprine. Cytotoxic drugs inhibit the immune response by killing dividing cells such as activated T cells. However, the killing is indiscriminate and other constantly dividing cells and their organs are affected, which causes toxic side effects. Immunosuppressive drugs prevent T cells from responding to signals correctly by inhibiting signal transduction pathways.

Question: What is an example of an immunosuppressive drug that prevents T cell activity by altering signal transduction pathways?

Justification: the specific name 'cyclosporin' is removed from the context, leaving no direct reference to any immunosuppressive drug that alters signal transduction pathways in T cells. Without naming a specific drug, the answer to the question becomes unknown.

Answer: The context states that immunosuppressive drugs prevent T cells from responding to signals by inhibiting signal transduction pathways. It provides examples of such drugs as methotrexate or azathioprine when used in conjunction with anti-inflammatory drugs. However, the passage distinguishes cytotoxic drugs (e.g., methotrexate, which kills dividing cells) from immunosuppressive drugs (e.g., azathioprine, which targets signal transduction). Since the question specifically asks for an immunosuppressive drug that alters signal transduction pathways, **azathioprine** is the correct example based on the context provided. \n\nAnswer: azathioprine"

Figure 10: Input-to-Trajectory Hallucination

Context: Teachers face several occupational hazards in their line of work, including occupational stress, which can negatively impact teachers' mental and physical health, productivity, and students' performance. Stress can be caused by organizational change, relationships with students, fellow teachers, and administrative personnel, working environment, expectations to substitute, long hours with a heavy workload, and inspections.

Question: What is something that teacher's are at a high risk for?

Justification: The sentence mentioning that teachers are at high risk for occupational burnout was removed. Therefore, the modified context no longer supports the original answer.

Answer: Okay, let's start by looking at the question: \"What is something that teachers are at a high risk for?\" The answer needs to come from the provided context.\n\nFirst, I'll read through the context again. It says that teachers face several occupational hazards, including occupational stress. Then it lists the negative impacts: mental and physical health problems, reduced productivity, and affecting students' performance. The causes of stress are mentioned as organizational change, relationships with students, colleagues, admin, work environment, expectations to substitute, long hours, heavy workload, and inspections.\n\nThe question asks for something teachers are at high risk for. The context clearly states occupational stress as a key hazard. So the main risk here is occupational stress. I should check if there's any other hazard mentioned besides stress. The context starts by listing occupational stress as one of the several hazards, but it doesn't specify others. All the other points are causes or effects, not additional hazards.\n\nSo the answer should be occupational stress because that's explicitly stated as an occupational hazard. The answer has to be pulled directly from the context provided, and in that case, it's mentioned. Other potential risks like health issues are outcomes, but the question is asking for what they are at risk for, which would be the stress itself. The context states that teachers face occupational stress as a hazard, so that's the primary answer here.\n\nTeachers are at a high risk for **occupational stress**. The context explicitly identifies occupational stress as a key hazard they face, caused by factors such as organizational changes, workplace relationships, heavy workloads, and inspections. This stress negatively impacts their mental and physical health, productivity, and students' performance. The answer is directly supported by the provided information. \n\n**Answer:** Occupational stress.

Figure 11: Input-to-Trajectory Hallucination

Context: In March 1896, after hearing of Wilhelm Röntgen\'s discovery of X-ray and X-ray imaging (radiography), Tesla proceeded to do his own experiments in X-ray imaging, developing a high energy single terminal vacuum tube of his own design that had no target electrode and that worked from the output of the Tesla Coil (the modern term for the phenomenon produced by this device is bremsstrahlung or braking radiation). In his research, Tesla devised several experimental setups to produce X-rays. Tesla held that, with his circuits, the "instrument will ... enable one to generate rays of much greater power than obtainable with ordinary apparatus.

Question: What were X-rays known as at the time?

Justification: The sentence mentioning 'Roentgen rays' was replaced with a more generic term 'rays' to remove any specific reference to the historical term for X-rays, thus ensuring the modified context does not support the original answer.

Answer: Okay, let's tackle this question step by step. The question is asking: \"What were X-rays known as at the time?\" The context provided is about Tesla's experiments in March 1896, following Wilhelm Röntgen's discovery.\n\nFirst, I need to recall the history. Wilhelm Röntgen discovered X-rays in 1895, and he named them \"X-rays\" because \"X\" stood for unknown, as he wasn't sure of their nature. At the time of Tesla's experiments in 1896, the term X-rays was already in use, right? But wait, sometimes discoveries have different names before they are officially named. Did Röntgen call them something else initially?\n\nWait, the context says \"after hearing of Wilhelm Röntgen's discovery of X-ray and X-ray imaging (radiography)\" so in the context, Röntgen's discovery is referred to as X-ray. So the term \"X-rays\" was already established by Röntgen himself. However, sometimes in different languages or regions, terms can vary. For example, in German, Röntgen called them X-Strahlen, and they might have been referred to as \"Röntgen rays,\" especially in Europe.\n\nBut the question is asking what they were known as at that specific time when Tesla was working on them in March 1896. The context mentions \"the modern term for the phenomenon produced by this device is bremsstrahlung or braking radiation.\" However, bremsstrahlung refers to the type of radiation, not the name of X-rays themselves.\n\nWait, but wait. The question says, \"What were X-rays known as at the time?\" So maybe they were known by another name then? Let me double-check. No, the context says Röntgen discovered X-rays, and the term used in the context is X-ray, so perhaps the answer is X-rays. But I need to confirm if there's any alternate term from the context provided that isn't relying on my external knowledge. The problem states that if there's no information in the context, the answer should be 'unknown.' But the context does mention \"X-ray\" as the term used by Röntgen. So according to the context given, the answer is \"X-rays.\"\n\nThe context provided states that Tesla was responding to Wilhelm Röntgen's discovery of \"X-ray and X-ray imaging (radiography).\" This explicitly uses the term \"X-ray\" to describe the phenomenon at the time of Tesla's experiments in 1896. There is no mention of an alternative name for X-rays in the context\n\nAnswer: **X-rays**.

Figure 12: Intra-Trajectory Hallucination

Context: Teachers face several occupational hazards in their line of work, including occupational stress, which can negatively impact teachers' mental and physical health, productivity, and students' performance. Stress can be caused by organizational change, relationships with students, fellow teachers, and administrative personnel, working environment, expectations to substitute, long hours with a heavy workload, and inspections.

Question: What is something that teacher's are at a high risk for?

Justification: The sentence mentioning that teachers are at high risk for occupational burnout was removed. Therefore, the modified context no longer supports the original answer.

Answer: Okay, let's start by looking at the question: \"What is something that teachers are at a high risk for?\" The answer needs to come from the provided context.\n\nFirst, I'll read through the context again. It says that teachers face several occupational hazards, including occupational stress. Then it lists the negative impacts: mental and physical health problems, reduced productivity, and affecting students' performance. The causes of stress are mentioned as organizational change, relationships with students, colleagues, admin, work environment, expectations to substitute, long hours, heavy workload, and inspections.\n\nThe question asks for something teachers are at high risk for. The context clearly states occupational stress as a key hazard. So the main risk here is occupational stress. I should check if there's any other hazard mentioned besides stress. The context starts by listing occupational stress as one of the several hazards, but it doesn't specify others. All the other points are causes or effects, not additional hazards.\n\nSo the answer should be occupational stress because that's explicitly stated as an occupational hazard. The answer has to be pulled directly from the context provided, and in that case, it's mentioned. Other potential risks like health issues are outcomes, but the question is asking for what they are at risk for, which would be the stress itself. The context states that teachers face occupational stress as a hazard, so that's the primary answer here.\n\nTeachers are at a high risk for **occupational stress**. The context explicitly identifies occupational stress as a key hazard they face, caused by factors such as organizational changes, workplace relationships, heavy workloads, and inspections. This stress negatively impacts their mental and physical health, productivity, and students' performance. The answer is directly supported by the provided information. \n\n**Answer:** Occupational stress.

Figure 13: Intra-Trajectory Hallucination

## G   LIMITATIONS AND FUTURE WORK

This study focuses on applying SQV to contextual reasoning tasks, aiming to address faithfulness and hallucination challenges in contextual question answering. Our findings validate the significant success of the thesis-antithesis-synthesis paradigm in these tasks. Moving forward, extending this paradigm to mathematical and procedural reasoning—where designing antithetical arguments for formulaic systems may require more sophisticated mechanisms—remains an open research direction. Furthermore, future work will focus on implementing thesis-antithesis-synthesis through latent reasoning frameworks, advancing the paradigm's adaptability and interpretability.

## H   THE USE OF LLMS

Large Language Models were used in this research solely as writing assistance tools to aid in language polishing and refinement.

