# OpenReview forum: "FaithThinker: Dialectical Reasoning for Noise-Robust LLMs"
_ICLR.cc/2026/Conference — Submitted to ICLR 2026_

### Official Review · Reviewer_JMUb · 2025-10-24

**Soundness:** 3
**Presentation:** 3
**Contribution:** 3
**Rating:** 6
**Confidence:** 4

**Summary:**

The paper presents FaithThinker, a new reasoning framework for large language models (LLMs) designed to improve contextual faithfulness—the model’s ability to remain consistent with retrieved evidence and avoid hallucinations, even in noisy or adversarial contexts. The framework is built on the Self-Questioning and Verification (SQV) paradigm, which encourages models to question, verify, and refine their reasoning steps to enhance accuracy and reliability. In addition, the study introduces a fine-tuning approach that integrates this reasoning ability directly into the model’s inherent capabilities, ensuring more faithful and robust reasoning performance.

**Strengths:**

The paper introduces a highly original and well-motivated framework, FaithThinker, that addresses a critical limitation in current reasoning systems—maintaining contextual faithfulness in the presence of noisy or adversarial contexts. Its central innovation, the Self-Questioning and Verification (SQV) paradigm, represents a creative and conceptually elegant integration of dialectical reasoning into LLM inference. Unlike existing structured reasoning frameworks such as Chain-of-Thought, Self-Refine, or Tree-of-Thought, SQV achieves verification within a single forward pass, effectively balancing reasoning depth with computational efficiency.

In terms of originality, the work stands out by redefining how reasoning reliability can be formalized and improved. The formulation of Input-to-Trajectory and Intra-Trajectory Hallucinations provides a new lens for understanding reasoning failures. The proposed SQV-Alignment method further extends this contribution by offering a practical and scalable approach to transferring reasoning capabilities from large to smaller models through adversarial context fine-tuning.

The quality of the work is strong, supported by comprehensive experiments across multiple model scales and benchmarks. Results consistently demonstrate substantial reductions in hallucination rates and improvements in efficiency, indicating both technical soundness and empirical robustness.

The clarity of the paper is commendable. The structure is logical, with motivating examples, clear mathematical formalization, and detailed illustrations (e.g., Figure 2) that make the methodology easy to follow.

Regarding significance, the work is highly relevant to both academic and applied research in reasoning, retrieval-augmented generation, and trustworthiness of LLMs. The proposed dialectical reasoning paradigm has potential to influence future directions in robust AI reasoning design and knowledge alignment, making the paper an important step toward self-corrective LLMs.

**Weaknesses:**

While the conceptual contribution is novel, there are a few areas that could be strengthened. First, the paper would benefit from more detailed ablation studies on the role of each SQV component (thesis, antithesis, verification, synthesis) across different model sizes to better isolate which stages contribute most to performance gains.

Second, the evaluation scope, though broad, focuses primarily on QA-style reasoning benchmarks. The paper could explore more complex multi-step reasoning domains (e.g., mathematical reasoning, scientific inference, or code understanding) to demonstrate broader generalization of SQV.

Third, while SQV-Alignment is introduced as a scalable fine-tuning approach, the paper does not provide computational cost comparisons or training efficiency metrics relative to standard fine-tuning methods. A clearer analysis of scalability and training stability would strengthen the claim of efficiency.

In addition, more models should be tested and validated to better demonstrate the generalizability of the proposed method and to provide broader and more comprehensive benchmark results for the study.

Finally, some theoretical parts (e.g., Equations 2–4) could benefit from tighter connections to empirical findings—for instance, showing how the probabilistic formulation directly informs prompt design or evaluation metrics.

**Questions:**

1. How does the SQV framework interact with retrieval quality? Specifically, can SQV detect when the retrieved evidence itself is misleading, and to what extent does it mitigate such failures without additional retrieval filtering?

2. Can the authors provide a more detailed breakdown of computational efficiency—both in terms of reasoning path length and wall-clock inference time—compared to iterative methods like Self-Refine?

3. How sensitive is SQV-Alignment to the choice of teacher model and noise type during adversarial fine-tuning? Would the framework generalize if the teacher model were significantly smaller or trained on a different domain?

4. Have the authors considered combining SQV with reinforcement learning or process supervision methods to further strengthen long-horizon reasoning? If so, what were the observed trade-offs?

5. For interpretability and reproducibility, could the authors share more qualitative examples that illustrate how SQV reasoning evolves differently from standard Chain-of-Thought under adversarial input conditions?

---

> ### Author Response · Authors · 2025-11-20
> **Response to Reviewer JMUb [Weakness 1,2]**
>
> Thank you for your thoughtful review and insightful comments. We hereby address your concerns below:
> - **W1**: the paper would benefit from more detailed ablation studies on the role of each SQV component (thesis, antithesis, verification, synthesis) across different model sizes to better isolate which stages contribute most to performance gains.
>
>  **A1**: Thank you for this valuable suggestion. We have conducted comprehensive ablation studies to isolate the contribution of each SQV component across different model scales.
>   - Expanded Ablation Studies: Building on the dialectical component ablation in Section 4.4, we now include detailed experiments that individually ablate the thesis, antithesis, and combined verification+synthesis stages. Here, we take the original ablation of the dialectical component as the ablation of antithesis. The prompt templates for these ablations are provided in Appendix B.1 of the revised manuscript.
>   - Methodology: For each ablation, we follow the identical training procedure as SQV-Alignment—using the same dataset, hyperparameters, and SFT configuration—while modifying only the prompt template to exclude the target component. This ensures that performance differences are solely attributable to the removed stage.
>
>   Key Findings (As displayed in the following table):
>   1. All components are essential: Ablating any stage degrades performance across all model sizes
>   2. Antithesis is most critical: Removing antithesis causes the largest performance drops (e.g., -32.0% on FaithEval-C for Qwen2.5-3B and -27.2% on ConFiQA for Deepseek-R1), confirming its role in generating diverse perspectives and reducing hallucinations.
>
>
>   | Method | FaithEval-U| FaithEval-I | FaithEval-C | ConFiQA | TriviaQA | NQ
>   | :--- |:--- | :--- | :--- | :--- | :--- | :--- |
>   | Qwen2.5-3B | |  |  |  | | |
>   | -w/o. A | 81.0  | 74.5  | 37.7 | 53.4 |57.9 | 41.8|
>   | -w/o. T | 86.4 | 74.9 | 58.3 | 60.4 | 57.9 | 44.4|
>   | -w/o. V,S| 85.6 | 74.2 | 48.3 | 68.2 | 77.1 | 49.4 |
>   | Deepseek-R1 |   |  |  |  | |  |
>   | -w/o. A | 75.6| 99.2 | 48.4 | 66.4 | 68.4| 55.2 |
>   | -w/o. T | 83.6| 99.8 | 51.2 | 69.8 | 72.9| 55.0 |
>   | -w/o. V,S| 81.6| 99.2 | 52.5| 67.0 | 77.6| 55.6 |
>
>   We will add the ablations and results in the revised manuscript.
>
> - **W2**: the evaluation scope, though broad, focuses primarily on QA-style reasoning benchmarks. The paper could explore more complex multi-step reasoning domains (e.g., mathematical reasoning, scientific inference, or code understanding) to demonstrate broader generalization of SQV.
>
>  **A2**: Thank you for the observation. Our work specifically targets context-dependent tasks (as detailed in Section 3.1) , where models must ground their reasoning in externally retrieved documents—a scenario inherently prone to noise (as the quality of such sources varies) . Following [1][2][3][4], we aim to **enhance context faithfulness** by improving structured reasoning patterns in precisely this setting.
>
>   Mathematical reasoning tasks fall under logical reasoning rather than context-dependent reasoning. These tasks present clean, self-contained problems without external retrieval, making them unsuitable for evaluating robustness to noisy contexts. Indeed, to the best of our knowledge, few benchmarks simulate realistic noise in mathematical reasoning, as such scenarios are uncommon in practice.
>
>   Our contribution addresses the critical, well-documented challenge of robust reasoning under noisy or adversarial contexts in context-dependent tasks (e.g. RAG), where FaithThinker effectively mitigates hallucinations. While extending structured reasoning improvements to math tasks is valuable, it would require developing novel noisy-context benchmarks for logical reasoning—a substantial endeavor we reserve for future work. Our current focus enables a thorough investigation of context faithfulness where it is most urgently needed [1][2][3][4].

---

> ### Author Response · Authors · 2025-11-20
> **Response to Reviewer JMUb [Weakness 3-5]**
>
> - **W3**: while SQV-Alignment is introduced as a scalable fine-tuning approach, the paper does not provide computational cost comparisons or training efficiency metrics relative to standard fine-tuning methods. A clearer analysis of scalability and training stability would strengthen the claim of efficiency.
>
>  **A3**: Thank you for your feedback. To address computational efficiency comparisons, we provide additional clarifications based on Appendix Table 4:
>   In our experiments, we evaluated two training strategies for SQV-Alignment—SFT-only and SFT+GRPO—and compared their computational costs and performance. The results demonstrate that under identical settings, the SFT-only approach achieves a 200× speedup over reinforcement learning-based alignment methods (0.09 vs. 20 GPU hours), outperforming SFT+GRPO in both efficiency and performance metrics (see Appendix C.2 for details).
>   For more detailed presentation of efficiency, we include training parameter percentages and GPU-hour requirements across model sizes:
>
>   | Model  | Trainable Parameters (%)| Training Cost (GPU Hours)|
>   | :--- |:--- | :--- |
>   | Qwen2.5-3B | 3.7 | 0.09 |
>   | Llama3.1-8B | 3.8 |  0.22|
>
> - **W4**: In addition, more models should be tested and validated to better demonstrate the generalizability of the proposed method and to provide broader and more comprehensive benchmark results for the study.
>
> **A4**: Thank you for the valuable feedback. In our study, we have conducted extensive evaluations across models of varying scales to rigorously validate the generalizability of the proposed SQV method, including:
>   - Small/Medium Models: Qwen2.5-3B-Instruct, Llama3.1-8B-Instruct
>   - Large Models: Qwen2.5-32B-Instruct, Qwen2.5-72B-Instruct (Appendix D.4)
>   - 100B+ Models: DeepSeek-R1
>
>   Across all tested model sizes, SQV consistently outperformed other structured reasoning patterns, demonstrating its generalizability and superiority.
>
> - **W5**: Finally, some theoretical parts (e.g., Equations 2–4) could benefit from tighter connections to empirical findings—for instance, showing how the probabilistic formulation directly informs prompt design or evaluation metrics.
>
> **A5**: We appreciate the suggestion to strengthen the connection between our theoretical framework and empirical findings. In Appendix A, we provide a probabilistic analysis from an ensemble learning perspective that directly motivates both our prompt design and evaluation methodology.
>
>   - Theoretical Analysis of Hallucination Reduction (Appendix A.2)
>     Our formal analysis demonstrates that the SQV framework achieves a lower hallucination rate compared to conventional Chain-of-Thought (CoT) approaches. Specifically, we show how the probabilistic formulation in Equation 4 systematically reduces the occurrence probabilities captured in Equations 2 and 3:
>     Let ε denote the hallucination probability at each reasoning step. We define the hallucination rates (aggregate probability of Equations 2 and 3) as:
>     - Chain-of-Thought (CoT):  $P_\mathrm{COT}({\text{Hal}}) = \epsilon$
>     - Our SQV framework: $P_\mathrm{SQV}({\text{Hal}}) = \epsilon^2$
>
>     **Theoretical Derivation**:
>     The SQV framework enables cross-path error cancellation through its adversarial structure. The correctness of final answers depends on three mutually exclusive scenarios:
>     1. Both Thesis and Antithesis incorrect: System likely produces incorrect output
>     2. One path correct, one incorrect: Verification mechanism weights correct path
>     3. Both paths correct: System guaranteed to produce correct output
>     Assuming the Verification phase correctly assesses statement reliability, and that Thesis (T) and Antithesis (A) errors are independent with $ε_T = ε_A = ε$, we derive:
>
>     $P_\mathrm{SQV}({\text{Hal}})$ = $P (\text{Hal}_T \cap \text{Hal}_A)$ = $\epsilon_T \cdot \epsilon_A = \epsilon ^2$
>
>     This establishes that $P_\mathrm{SQV}({\text{Hal}})$ = $\epsilon ^2 < \epsilon = P_\mathrm{COT}({\text{Hal}})$ for 0 < $\epsilon$ < 1.
>
>     Furthermore, we demonstrate $P_\mathrm{SQV}({\text{Hal}}) < P_\mathrm{SC}({\text{Hal}})$ when compared to Self-Consistency (SC) approaches. Detailed derivations are available in Appendix A.2.
>   - Ensemble Learning Interpretation (Appendix A.1)
>     We further analyze SQV through the lens of ensemble learning, providing insights into its effectiveness and suggesting directions for future research. The framework conceptualizes:
>     - Base learners: Thesis and Antithesis generators producing multiple reasoning paths
>     - Ensemble learner: Verification and Synthesis stages performing meta-reasoning
>     The counterfactual nature of SQV's base learners - specifically the opposing pair ($\theta_T$,$\theta_A$) - inherently promotes diversity among reasoning paths. This architectural characteristic aligns with established ensemble learning principles [5], where increased diversity among base learners leads to improved generalization performance.

---

> ### Author Response · Authors · 2025-11-20
> **Response to Reviewer JMUb [Questions 1-3]**
>
> - **Q1**: How does the SQV framework interact with retrieval quality? Specifically, can SQV detect when the retrieved evidence itself is misleading, and to what extent does it mitigate such failures without additional retrieval filtering?
>
> **A1**: The overall objective of our work in noisy RAG contexts, is faithfulness to the provided context—not factual correctness based on world knowledge, following [1][6][7]. Its design focuses on detecting and mitigating errors arising from misleading or noisy retrieved evidence through context-constrained reasoning, without relying on additional retrieval filtering. As detailed in Appendix D.1.1, we systematically outline the requirements for addressing the three critical noisy-context scenarios with the benchmark framework established in [1]:
>  - **Unanswerable Context.** It occurs when the provided context, while containing relevant information, lacks sufficient details to derive a correct answer. Answerability is strictly context-dependent and independent of the question's inherent solvability. When processing unanswerable contexts, LLMs should **explicitly acknowledge the lack of sufficient information by responding with "Unanswerable" or equivalent indication, rather than generating speculative answers that disregard contextual constraints**.
>  - **Inconsistent Context.** Feature multiple documents that present conflicting answers to the same query, simulating real-world scenarios where retrieved information comes from sources with varying reliability. When processing inconsistent contexts, LLMs should **explicitly acknowledge the contradiction within the contexts, rather than generating speculative answers that disregard contextual constraints**.
>  - **Counterfactual Context.** Contain assertions that violate fundamental world knowledge (e.g., "water freezes at 100°C," "wood exhibits magnetic properties," or "carbon dioxide constitutes the most abundant atmospheric greenhouse gas"). When processing counterfactual contexts, LLMs should **strictly adhere to and reason from the provided context, while suppressing contradictory parametric knowledge that would normally lead to factually correct responses.**
>
> - **Q2**: Can the authors provide a more detailed breakdown of computational efficiency—both in terms of reasoning path length and wall-clock inference time—compared to iterative methods like Self-Refine?
>
> **A2**: To quantify the inference efficiency, we have included wall-clock and GPU hours for inference of the methods in Appendix Table 5 and displayed the results in Figure 6 in the updated pdf.
>
> - **Q3**: How sensitive is SQV-Alignment to the choice of teacher model and noise type during adversarial fine-tuning? Would the framework generalize if the teacher model were significantly smaller or trained on a different domain?
>
> **A3**: Thank you for the valuable feedback. To evaluate SQV-Alignment's sensitivity to teacher models and noise types, along with its generalization capabilities, we conducted experiments using PubMedQA, a biomedical QA dataset representing a different domain. We replicated our original training procedure: first generating adversarially perturbed text for PubMedQA using the same noise types, then applying identical data filtering protocols to create a curated subset of 1,000 adversarial examples. We fine-tuned Qwen2.5-3B on this dataset while maintaining identical hyperparameters and optimization strategies for fair comparison.
>
>   As displayed in the following table, the results demonstrated only marginal performance degradation despite the domain shift, with significantly outperforming baseline methods in noisy contexts. This further confirms robust generalization of SQV-Alignment.
>
>   | Method | FaithEval-U| FaithEval-I | FaithEval-C | ConFiQA | TriviaQA | NQ
>   | :--- |:--- | :--- | :--- | :--- | :--- | :--- |
>   | Pubmed | 92.2  | 94.4 | 68.4 | 78.6 | 83.6 | 55.2 |

---

> ### Author Response · Authors · 2025-11-20
> **Response to Reviewer JMUb [Questions 4]**
>
> - **Q4**: Have the authors considered combining SQV with reinforcement learning or process supervision methods to further strengthen long-horizon reasoning? If so, what were the observed trade-offs?
>
>  **A4**: Yes, we explored combining SQV with reinforcement learning, specifically using the GRPO algorithm, to enhance long-horizon reasoning. Detailed analysis and experiments are provided in Appendix C.2.
>   While reinforcement learning has emerged as a key approach in LLM post-training, aligning models with human preferences and enhancing their reasoning abilities, however, in this scenario, we find SFT serves as a more efficient optimization method for SQV alignment through experiments. In practice, we utilize the GRPO algorithm for model refinement, designing a composite reward function that integrates accuracy rewards and format rewards(Appendix C.2.1). Experiments conducted on Qwen2.5 3B (Table 4) show that after GRPO training, the model's success rate in generating correct SQV formats (quantified as SQV Format Ratio) declines significantly from 65.2\% to 10.5\%, with consistent performance degradation observed across all datasets. Figure 6 illustrates a representative failure case where the model after GRPO generates incorrect SQV formatting. Concurrently, GRPO introduces substantial computational overhead, increasing training time from SFT's efficient 0.09 hours to 20 hours. We hypothesize this performance deterioration stems from the inherent tension between SQV's precise formatting requirements and RL's exploration-driven paradigm [8]: outcome-based rewards fail to effectively balance granular format constraints against accuracy optimization in extended reasoning paths, ultimately impairing the model's instruction-following capability for SQV generation. Notably, this experiment further validates the critical role of SQV-formatted reasoning in contextual faithfulness. For future work, we plan to explore integrating SQV's reasoning structure with intrinsic rewards and process supervision methods, aiming to enable correct SQV reasoning through RL while improving overall performance.

---

> ### Author Response · Authors · 2025-11-20
> **Response to Reviewer JMUb [Questions 5, Part 1]**
>
> - **Q5**: For interpretability and reproducibility, could the authors share more qualitative examples that illustrate how SQV reasoning evolves differently from standard Chain-of-Thought under adversarial input conditions?
>
>  **A5-part1**: Thanks for the feedback. To enhance interpretability, we provide concrete examples contrasting SQV reasoning and standard Chain-of-Thought under adversarial conditions below:
>   - Example 1
>     - Question: What space station supported three manned missions in 1973-1974?
>     - Noisy Context: Apollo ran from 1961 to 1972, and was supported by the two-man Gemini program which ran concurrently with it from 1962 to 1966. Gemini missions developed some of the space travel techniques that were necessary for the success of the Apollo missions. Apollo used Saturn family rockets as launch vehicles. Apollo/Saturn vehicles were also used for an Apollo Applications Program, which included various other missions and projects.
>     - Justification: The sentence explicitly mentioning 'Skylab' and its association with three manned missions in 1973-74 was removed, and replaced with a vague reference to 'various other missions and projects'. This alteration removes any detail that could directly answer the question about the specific space station involved in the missions in 1973-1974.
>     - standard COT: \<think\>\nOkay, let's see. The question is asking which space station supported three manned missions in 1973-1974. The context provided mentions the Apollo program running from 1961 to 1972 and the Gemini program from 1962 to 1966. It also talks about the Apollo Applications Program using Apollo/Saturn vehicles for other missions.\n\nHmm, I remember that after the Apollo program, NASA used leftover hardware for other projects. The Apollo Applications Program included Skylab, right? Skylab was the United States' first space station. It was launched in 1973, and there were three manned missions to Skylab: Skylab 2, 3, and 4. These missions took place in 1973 and 1974. The context mentions the Apollo Applications Program, which Skylab was part of. So even though Skylab isn't directly named in the context, the Apollo Applications Program is referenced, and Skylab is the known project under that program. Therefore, the answer should be Skylab.\n\</think\>\n\nThe Apollo Applications Program, mentioned in the context, included missions such as Skylab, the first U.S. space station. Skylab hosted three manned missions (Skylab 2, 3, and 4) between 1973 and 1974, utilizing Apollo/Saturn hardware. \n\nAnswer: Skylab.
>     - SQV: The space station that supported three manned missions in 1973–1974 was Skylab. It was part of the Apollo Applications Program, which utilized Apollo/Saturn vehicles as stated in the context. The three crewed Skylab missions (Skylab 2, 3, and 4) occurred during this period.  \n\<sub-question\>  \n1. Does the context explicitly name the space station associated with the Apollo Applications Program?  \n\</sub-question\>  \n\<sub-answer\>  \n1. No, the context only mentions the Apollo Applications Program and its use of Apollo/Saturn vehicles but does not name Skylab.  \n\</sub-answer\>  \n\<certainty\>  \n1. **Counterfactual Question:** What if the Apollo Applications Program did not include Skylab?  \n2. **Re-examination:** The context states the Apollo Applications Program included "various other missions and projects" using Apollo/Saturn hardware. Skylab is a known historical component of this program, though not explicitly named here.  \n3. **Evidence:** While the context does not provide the name "Skylab," historical records confirm Skylab was the only U.S. space station operating in 1973–1974 and was part of the Apollo Applications Program.  \n4. **Conclusion:** Despite lacking explicit naming in the context, Skylab aligns logically with the described program. However, strictly adhering to the given context, the answer would be "unknown," but real-world knowledge fills this gap. The answer is inferred as Skylab with medium certainty due to contextual hints.  \n\</certainty\> \n\n**Final Answer**:  \nunknown.

---

> ### Author Response · Authors · 2025-11-20
> **Response to Reviewer JMUb [Questions 5, Part 2 & References]**
>
> **A5-part2**:
> - Example 2
>     - Question: What kind of sending technology is being used to protect tribal lands in the Amazon?
>     - Noisy Context: Indigenous tribes of the Amazon basin are employing technology to safeguard their tribal lands from commercial interests. Using handheld GPS devices and programs like Google Earth, members of the Trio Tribe, who live in the rainforests of southern Suriname, map out their ancestral lands to help strengthen their territorial claims. Currently, most tribes in the Amazon do not have clearly defined boundaries, making it easier for commercial ventures to target their territories.
>     - Justification: The modified context removes all mentions of 'remote sensing' and introduces a general statement about tribes using technology without specifying the type. This change ensures that the specific technology used (remote sensing) is not revealed, thereby not supporting the original answer.
>     - standard COT: \<think\>\nOkay, let\'s see. The question is asking what kind of "sending technology" is being used by indigenous tribes in the Amazon to protect their lands. Wait, that might be a typo. The context mentions GPS devices and Google Earth. The user wrote "sending technology," but maybe they meant "sensing technology"? Because GPS and mapping software like Google Earth are technologies that sense or map the environment.\n\nLooking at the context again: It says the Trio Tribe uses handheld GPS devices and Google Earth to map their lands. These tools help them create maps to support their territorial claims. So the answer should be GPS and Google Earth. The term "sending technology" doesn\'t quite fit here. Maybe it\'s a mistake, and the intended term was "sensing" or "mapping technology." But according to the context provided, the answer explicitly mentions handheld GPS devices and Google Earth. The question might have a typo, but based on the information given, the technologies used are GPS and Google Earth. The answer should list those two as the technologies being employed.\n\</think\>\n\nThe indigenous tribes are using handheld GPS devices and programs like Google Earth to map and protect their tribal lands. These technologies help in documenting ancestral territories to support territorial claims against commercial interests.\n\nAnswer: handheld GPS devices and Google Earth.
>     - SQV: 1. \<sub-question\>  \n   What specific technologies are mentioned in the context for mapping tribal lands?  \n   \</sub-question\>  \n   \<sub-answer\>  \n   The context specifies handheld GPS devices and Google Earth as technologies used by the Trio Tribe.  \n   \</sub-answer\>  \n\n2. \<sub-question\>  \n   Does the context mention any technology used explicitly for "sending" data (e.g., transmitting information to external parties)?  \n   \</sub-question\>  \n   \<sub-answer\>  \n   No. The context focuses on mapping tools, not communication or data transmission technologies.  \n   \</sub-answer\>  \n\n3. \<certainty\>  \n   1. **Counterfactual Question**: If the tribes used satellite internet to transmit mapped data, would that fall under "sending technology"?  \n   2. **Re-examination**: The context does not describe transmission methods, only mapping tools (GPS, Google Earth).  \n   3. **Evidence**: The passage emphasizes mapping to "strengthen territorial claims" but does not discuss how data is shared or transmitted.  \n   4. **Conclusion**: The term "sending technology" might be a misinterpretation (e.g., "sensing" or "mapping" intended). Based on the context, no "sending" technology is mentioned; the focus is on mapping tools.  \n   </certainty>  \n\n**Final Answer**:  \nThe technology explicitly mentioned for protecting tribal lands includes handheld GPS devices and Google Earth. If "sending" refers to data transmission, the answer is **unknown** per the context. If a terminology error occurred (e.g., "sensing" or "mapping"), the answer aligns with GPS and Google Earth.
>
>
> **References**:
>
> [1] FaithEval: Can Your Language Model Stay Faithful to Context, Even If "The Moon is Made of Marshmallows" (Ming et al., ICLR 2025)
>
> [2] Evaluating Correctness and Faithfulness of Instruction-Following Models for Question Answering (Adlakha et al., TACL 2024)
>
> [3] Benchmarking large language models in retrieval-augmented generation (Chen et al., AAAI 2024)
>
> [4] Context-DPO: Aligning Language Models for Context-Faithfulnes (Bi et al., ACL 2025)
>
> [5] A survey of ensemble learning: Concepts, algorithms, applications, and prospects (Zhou, Z.H., et al., "IEEE Transactions on Knowledge and Data Engineering (2022)).
>
> [6] A Survey on Hallucination in Large Language Models: Principles, Taxonomy, Challenges, and Open Questions (Huang et al., ACM Trans. Inf. Syst)
>
> [7] Cognitive mirage: A review of hallucinations in large language models (Ye et al.)
>
> [8] Guiding Pretraining in Reinforcement Learning with Large Language Models (Du et al., ICML 2023)

---

> > ### Author Response · Authors · 2025-11-27
> >
> > Dear Reviewer,
> >
> > I hope this message finds you well. As the discussion period is nearing its end with less than one week remaining, I wanted to ensure that we have addressed all your concerns satisfactorily. If there are any additional points or feedback you'd like us to consider, please let us know.
> >
> > Thank you for your time and effort in reviewing our paper.

---

### Official Review · Reviewer_7zNZ · 2025-10-30

**Soundness:** 3
**Presentation:** 3
**Contribution:** 3
**Rating:** 6
**Confidence:** 3

**Summary:**

This paper introduces FaithThinker which is a reasoning framework aiming at improving the contextual faithfulness of the LLMs under noisy contexts. The core pattern which is Self questioning and verification is broadly a loop of thesis, antithesis, verification and finally sinthesis in a single forward pass to detect reasoning errors.
The authors perform a good study by defining two intraprocess hallucination types, input to trajectory and intra trajectory, with binary predicates over reasoning traces. They also propose SQV-alignment, a lora based sft process where training data is generated by strong teacher model to transfer the SQV pattern to smaller students. They experiment and show significant gains on multiple datasets and also show lower hallucination rates compared to CoT, SC, Self-refine methods.

**Strengths:**

1) Most important strength of the paper is the clear problem formulation of intra-process halluincations, where they explicitly distinguish hallucinations relative to context vs within trajectory and formalize both with simple indicators Hcr, and Hrr. This is a useful conceptual lens beyond final-answer hallucinations
2) The single pass dialectical control which is via the 4 stage SQV embeds verficiation into each micro steps, promising better error containment without the cost of iterative loop, the formulation and figure 2 are clear.
3) Data efficient alignment process, that is well motivated by using a teacher to synthesize SQV and adverserial contexts, Lora also keeps the compute low, making this widely useable.

The paper also shows minor but important contributions like token length differences and shows large savings versus multi sample methods.

**Weaknesses:**

1) Current statistical rigor is insufficient, inference is run twice and averaged, no confidence intervals. A statement of statistical significance appears only for the entropy analysis without test specifications or p values.
2) Ablations compare “w/o SQV format alignment” and “w/o dialectical components,” but they still rely on format-specific supervision. There is no variant that keeps identical prompts/format while disabling only the verification decision.
3) The set includes CoT, Self-Consistency, ToT, and Self-Refine, but no retrieval centric methods like chain of verification, self-rag with critique. this makes it hard to place SQV againsts state of the art approaches.

**Questions:**

1) In appendix E hallucination rate is defined as Nn ​ /N, while accuracy is 𝑁𝑝 / 𝑁  and  since 𝑁 𝑝 + 𝑁 𝑛 = 𝑁 doesnt the hallucination metric collapse into error rate, these two are perfectly anti-correlated (Hal = 1 - Acc), so this doesnt seem to measure faitfulness independent of correctness.
2) The much re-iterated claim of 4x shorter than iterative methods seems an overstretch, looking at table 5, self-refine is 1957 vs SQV is 596, this makes it more of like 3x (it holds for self consistency - but the abstract mentions self-refine which is untrue) for reference snippet from the abstract "4× shorter than iterative methods such as Self-Refine" this is false.
3) Typo: Line 338 says authors used Qwen2.5-3B and then "for brevity" they say on line 340 Qwen2.5-7B, minor point for correction.
4) Can you also report wall-clock and GPU hours for inference to quantify single pass benefits apart from token length

---

> ### Author Response · Authors · 2025-11-20
> **Response to Reviewer 7zNZ [Weakness 1-3 & Questions 1]**
>
> Thank you for your thoughtful review and insightful comments. We hereby address your concerns below:
> - **W1**: Current statistical rigor is insufficient, inference is run twice and averaged, no confidence intervals. A statement of statistical significance appears only for the entropy analysis without test specifications or p values.
>
>  **A1**: We thank the reviewer for highlighting the need for enhanced statistical rigor. In our updated manuscript, we have incorporated 95% confidence intervals across all the metrics in Table 1 to quantify measurement variability.
>
> - **W2**: Ablations compare “w/o SQV format alignment” and “w/o dialectical components,” but they still rely on format-specific supervision. There is no variant that keeps identical prompts/format while disabling only the verification decision.
>
>  **A2**: Thanks for the valuable feedback. To address the concern, we appended a complementary ablation in which the verification decision is completely disabled while the prompt format remain unchanged. We conducted experiments on Qwen2.5-3B
>   and Deepseek-R1. As demonstrated in the following table, the absence of the verification decision
>   resulted in consistent performance significant degradation across all evaluated datasets, with Qwen2.5-
>   3B and Deepseek-R1 experiencing a 18.2% and 24.1% decrease in average accuracy respectively. These results confirm that the self-questioning-verification mechanism, instead of the format alignment itself, is the primary driver of SQV’s hallucination-mitigation effect.
>
>   | Method | FaithEval-U| FaithEval-I | FaithEval-C | ConFiQA | TriviaQA | NQ
>   | :--- |:--- | :--- | :--- | :--- | :--- | :--- |
>   | Qwen2.5-3B | |  |  |  | | |
>   | -w/o. (T,A,V,S) | 73.8 | 75.0 |  50.6 | 53.6 | 64.8| 48.8 |
>   | Deepseek-R1 |   |  |  |  | |  |
>   | -w/o. (T,A,V,S) | 59.4| 87.7 | 46.2 | 59.6 | 65.6| 54.2 |
>
>
> - **W3**: The set includes CoT, Self-Consistency, ToT, and Self-Refine, but no retrieval centric methods like chain of verification, self-rag with critique. this makes it hard to place SQV againsts state of the art approaches.
>
>  **A3**: Thanks for the valuable feedback. To address these concerns, we extended our comparison to include Chain-of-Verification (CoVe) and Self-RAG with Critique. We evaluated the methods on Llama-3.1-8B. The results, shown in the following table, reveal that both Self-RAG and CoVe performs poorly, especially under noisy contexts, displaying significantly compromised robustness. Self-RAG with Critique, lacking the reasoning ability, falls markedly behind the reasoning-centric baselines, whereas CoVe narrows but does not eliminate the gap to SQV, thereby situating our method more clearly within the current retrieval-augmented landscape.
>
>   | Method | FaithEval-U| FaithEval-I | FaithEval-C | ConFiQA | TriviaQA | NQ
>   | :--- |:--- | :--- | :--- | :--- | :--- | :--- |
>   | $\text{SQV}_\text{alignment}$ |82.6 | 97.6 | 63.3 | 76.4 | 82.9| 59.3|
>   | Self-RAG |  49.0 | 24.0 | 45.8 | 42.0 | 64.2| 38.4|
>   | CoVe |  53.0 | 41.0 | 60.8 | 69.4 | 72.0| 51.2  |
>
>
> - **Q1**: In appendix E hallucination rate is defined as Nn  /N, while accuracy is 𝑁𝑝 / 𝑁 and since 𝑁 𝑝 + 𝑁 𝑛 = 𝑁 doesnt the hallucination metric collapse into error rate, these two are perfectly anti-correlated (Hal = 1 - Acc), so this doesnt seem to measure faitfulness independent of correctness.
>
> **A1**: Thank you for the valuable feedback, we hereby address your concerns below:
>
>  First, we added experiments to measure faitfulness. As outlined in Section 3.1, two types of hallucinations—Input-to-Trajectory Hallucination and Intra-Trajectory Hallucination—compromise the faithfulness of the model. Currently, the main methods for evaluating the quality of reasoning trajectories include Rule-based matching, Intrinsic properties, and Neural evaluator models [1]. To comprehensively assess the faithfulness hallucination in reasoning steps, we expanded our evaluation framework beyond intrinsic metrics (top-10% group entropy) to include neural evaluation. Specifically, we employed LLM-as-a-Judge, a widely accepted approach for evaluating long-form reasoning quality [2][3][4]. We designed specialized prompts for DeepSeek-V3.1 to assess both input-to-trajectory hallucination (consistency with inputs) and intra-trajectory hallucination (internal logical coherence). For each sample, we conducted multiple independent evaluations and averaged the scores to ensure reliability. The results (In the updated pdf Figure 4) demonstrate that SQV reasoning significantly reduces both hallucination types compared to baseline chain-of-thought, yielding higher-quality reasoning trajectories and thus enhancing the faithfulness of the model.
>
>  Second, to improve clarity and presentation, we have updated Table 1 in the revised pdf to include only accuracy rates.

---

> ### Author Response · Authors · 2025-11-20
> **Response to Reviewer 7zNZ [Questions 2-4 & References]**
>
> - **Q2**: The much re-iterated claim of 4x shorter than iterative methods seems an overstretch, looking at table 5, self-refine is 1957 vs SQV is 596, this makes it more of like 3x (it holds for self consistency - but the abstract mentions self-refine which is untrue) for reference snippet from the abstract "4× shorter than iterative methods such as Self-Refine" this is false.
>
>  **A2**: Thank you for the valuable feedback. We have corrected this statement to accurately reflect the 3× reduction compared to self-refine in the updated pdf.
>
> - **Q3**: Typo: Line 338 says authors used Qwen2.5-3B and then "for brevity" they say on line 340 Qwen2.5-7B, minor point for correction.
>
>  **A3**: Thank you for the valuable feedback. We have corrected this statement in the updated pdf.
>
> - **Q4**: Can you also report wall-clock and GPU hours for inference to quantify single pass benefits apart from token length
>
>  **A4**: Thank you for the valuable feedback. To address the concern, we have displayed the results in Figure 6 and reported the results in Appendix Table 5 in the updated pdf.
>
> **References**:
>
> [1] Evaluating Step-by-step Reasoning Traces: A Survey (Lee et al.)
>
> [2] A Chain-of-Thought Is as Strong as Its Weakest Link: A Benchmark for Verifiers of Reasoning Chains (Jacovi et al., ACL 2024)
>
> [3] Synchronous Faithfulness Monitoring for Trustworthy
> Retrieval-Augmented Generation (Wu et al., EMNLP 2024)
>
> [4] RAGTruth: A Hallucination Corpus for Developing Trustworthy Retrieval-Augmented Language Models (Niu et al.)

---

> > ### Author Response · Authors · 2025-11-27
> >
> > Dear Reviewer,
> >
> > I hope this message finds you well. As the discussion period is nearing its end with less than one week remaining, I wanted to ensure that we have addressed all your concerns satisfactorily. If there are any additional points or feedback you'd like us to consider, please let us know.
> >
> > Thank you for your time and effort in reviewing our paper.

---

### Official Review · Reviewer_u4ka · 2025-11-02

**Soundness:** 2
**Presentation:** 3
**Contribution:** 2
**Rating:** 4
**Confidence:** 3

**Summary:**

FaithThinker introduces Self-Questioning and Verification (SQV), a reasoning paradigm inspired by dialectical thinking, to enhance contextual faithfulness and robustness in large language models (LLMs). Traditional reasoning methods (e.g., Chain-of-Thought or Self-Refine) suffer from error cascades when input contexts are noisy or adversarial. SQV mitigates this by embedding questioning, verification, and refinement at each reasoning step, enabling single-pass correction without iterative loops.

**Strengths:**

1. This work embeds the dialectical self-critique within a single reasoning pass.
2. The work shows consistent improvements across multiple models and datasets.
3. This work enables small models to gain reasoning robustness from large teacher models without high RL cost.

**Weaknesses:**

1. The authors claim "structured reasoning under noisy context," but do not test on the math reasoning task.
2. The reliance on a strong SQV teacher model seems like a type of knowledge distillation.
3. Since LLMs process all steps, there is no analysis on whether each step can introduce new hallucinations, such as SQV turns correct steps to incorrect steps.

**Questions:**

See weaknesses.

---

> ### Author Response · Authors · 2025-11-20
> **Response to Reviewer u4ka [Weakness 1,2]**
>
> Thank you for your thoughtful review and insightful comments. We hereby address your concerns below:
> - **W1**: The authors claim "structured reasoning under noisy context," but do not test on the math reasoning task.
>
>  **A1**: Thank you for the observation. Our work specifically targets context-dependent tasks (as detailed in Section 3.1) , where models must ground their reasoning in externally retrieved documents—a scenario inherently prone to noise (as the quality of such sources varies) . Following [1][2][3][4], we aim to **enhance context faithfulness** by improving structured reasoning patterns in precisely this setting.
>
>   Mathematical reasoning tasks fall under logical reasoning rather than context-dependent reasoning. These tasks present clean, self-contained problems without external retrieval, making them unsuitable for evaluating robustness to noisy contexts. Indeed, to the best of our knowledge, few benchmarks simulate realistic noise in mathematical reasoning, as such scenarios are uncommon in practice.
>
>   Our contribution addresses the critical, well-documented challenge of robust reasoning under noisy or adversarial contexts in context-dependent tasks (e.g. RAG), where FaithThinker effectively mitigates hallucinations. While extending structured reasoning improvements to math tasks is valuable, it would require developing novel noisy-context benchmarks for logical reasoning—a substantial endeavor we reserve for future work. Our current focus enables a thorough investigation of context faithfulness where it is most urgently needed [1][2][3][4].
>
> - **W2**: The reliance on a strong SQV teacher model seems like a type of knowledge distillation.
>
>  **A2**: Indeed, this is a knowledge distillation approach. Smaller models lack the robust instruction-following capabilities of larger models (detailed in  Appendix C.1 ), necessitating distillation to acquire the SQV reasoning ability—a process we term SQV-Alignment. We strengthen this alignment through adversarial training data augmentation, which is why we designate the overall method as Adversarial Training.

---

> ### Author Response · Authors · 2025-11-20
> **Response to Reviewer u4ka [Weakness 3 & References]**
>
> - **W3**: Since LLMs process all steps, there is no analysis on whether each step can introduce new hallucinations, such as SQV turns correct steps to incorrect steps.
>
>  **A3**: We appreciate this important concern regarding potential hallucination propagation in multi-step SQV processing.
>   On the one hand, the SQV pattern can introduce new hallucinations, particularly format hallucinations. As detailed in Appendix C.1, small language models exhibit limited SQVzerorobustness under noisy conditions. Rather than performing authentic SQV reasoning, they frequently engage in verbatim replication of instructional phrases within \<certain\> tags (Figure 5), essentially parroting the template instead of generating genuine verification steps. This phenomenon constitutes a format-specific hallucination unique to SQV pattern reasoning. To quantify this, we introduced the SQV Format Ratio—a rule-based metric measuring adherence to the correct SQV structure. Table 3 in Appendix demonstrates that after SQV-Alignment, this format hallucination decreases substantially, with corresponding performance improvements in noisy contexts, validating the effectiveness of SQV-Alignment strategy.
>   On the other hand, regarding your specific concern about "turning correct steps into incorrect steps," such deviations would inevitably conflict with the original context and fall under our defined category of input-to-trajectory hallucination. To comprehensively assess the hallucination in reasoning steps, we expanded our evaluation framework beyond intrinsic metrics (top-10% group entropy) to include neural evaluation. Specifically, we employed LLM-as-a-Judge, a widely accepted approach for evaluating long-form reasoning quality ([5][6][7]). We designed specialized prompts for DeepSeek-V3.1 to assess both input-to-trajectory hallucination (consistency with inputs) and intra-trajectory hallucination (internal logical coherence). For each sample, we conducted multiple independent evaluations and averaged the scores to ensure reliability. The results (in the updated pdf Figure 4) demonstrate that SQV reasoning significantly reduces both hallucination types compared to baseline chain-of-thought, yielding higher-quality reasoning trajectories. This suggests that while SQV introduces format complexity, its verification mechanisms actually mitigate rather than exacerbate substantive hallucinations.
>   We will further strengthen these analyses in the final manuscript and thank you for this constructive suggestion.
>
> **References**:
>
> [1] FaithEval: Can Your Language Model Stay Faithful to Context, Even If "The Moon is Made of Marshmallows" (Ming et al., ICLR 2025)
>
> [2] Evaluating Correctness and Faithfulness of Instruction-Following Models for Question Answering (Adlakha et al., TACL 2024)
>
> [3] Benchmarking large language models in retrieval-augmented generation (Chen et al., AAAI 2024)
>
> [4] Context-DPO: Aligning Language Models for Context-Faithfulnes (Bi et al., ACL 2025)
>
> [5] A Chain-of-Thought Is as Strong as Its Weakest Link: A Benchmark for Verifiers of Reasoning Chains (Jacovi et al., ACL 2024)
>
> [6] Synchronous Faithfulness Monitoring for Trustworthy Retrieval-Augmented Generation (Wu et al., EMNLP 2024)
>
> [7] RAGTruth: A Hallucination Corpus for Developing Trustworthy Retrieval-Augmented Language Models (Niu et al.)

---

> > ### Author Response · Authors · 2025-11-27
> >
> > Dear Reviewer,
> >
> > I hope this message finds you well. As the discussion period is nearing its end with less than one week remaining, I wanted to ensure that we have addressed all your concerns satisfactorily. If there are any additional points or feedback you'd like us to consider, please let us know.
> >
> > Thank you for your time and effort in reviewing our paper.

---

### Official Review · Reviewer_ckNu · 2025-11-02

**Soundness:** 3
**Presentation:** 2
**Contribution:** 3
**Rating:** 2
**Confidence:** 5

**Summary:**

This paper introduces a prompting procedure based on self-questioning for enhancing the faithfulness of reasoning models to the provided context. This leads to notable improvements on QA benchmarks with noisy contexts (FaithEval). The paper also shows how reasoning traces from stronger models can be distilled into smaller ones.

**Strengths:**

- The proposed method leads to notable improvements over CoT and other baselines, across a range of models
- The ablation study suggests that the dialectical approach of self-questioning and verification does in fact lead to strong improvements (at least on the tasks considered).

**Weaknesses:**

- The presentation of the paper could be improved. It's not clear if the unnecessary math introduced for the technique adds any insight (Eqs 1-4). The key idea is an improved prompting technique, but its presented as something more complex.
- The concepts of input-to-trajectory hallucination and intra-trajectory hallucination are discussed at length, but there is no empirical evidence about how prevalent these are, and whether the SQV technique actually reduces it. There is pretty much no analysis of what the SQV prompt does in practice for the problems studied here.
- There are terms used throughout like top-10% group entropy, token efficiency without any explanation in the main paper. Looking at the appendix these are actually simple ideas (e.g., inverse of the response length) -- so why not directly use simple terminology?
- I find it very strange that the main table presents two metrics hallucination and accuracy, which always sum up to 1. There is no need to present two metrics -- this is like reporting both accuracy and error rates together.
- At a high-level, the paper fails to make it clear what should be the overall objective in cases where the RAG context is noisy. Should the model rely on its internal knowledge in this case, or should it identify the problems in the context and report it? What is being measured by the benchmarks used here?
- Some important citations are missing, e.g., an ICLR 2025 paper [1].

[1] Huang, Yukun, et al. "To Trust or Not to Trust? Enhancing Large Language Models' Situated Faithfulness to External Contexts." arXiv preprint arXiv:2410.14675 (2024).

**Questions:**

See weaknesses.

---

> ### Author Response · Authors · 2025-11-20
> **Response to Reviewer ckNu [Weakness 1]**
>
> Thank you for your thoughtful review and insightful comments. We hereby address your concerns below:
> - **W1**: The presentation of the paper could be improved. It's not clear if the unnecessary math introduced for the technique adds any insight (Eqs 1-4). The key idea is an improved prompting technique, but its presented as something more complex.
>
>  **A1**: First, Equation 1 formally defines the context-based reasoning task, which serves two critical purposes: (1) it establishes a precise task formulation that helps readers intuitively grasp the operational setting of our method, and (2) it lays the groundwork for analyzing how textual noise induces hallucinations, creating the "context" for the subsequent hallucination characterization.
>
>   Second, Equations 2-4 form the theoretical backbone that bridges our prompting strategy with empirical observations. In Appendix A, we provide theoretical proofs demonstrating how SQV's reasoning pattern (Equation 4) systematically reduces hallucination probabilities (as defined in Equations 2-3). This formal connection makes our method more interpretable. In Appendix A, we provide a probabilistic analysis from an ensemble learning perspective that directly motivates both our prompt design and evaluation methodology.
>
>   - Theoretical Analysis of Hallucination Reduction (Appendix A.2)
>     Our formal analysis demonstrates that the SQV framework achieves a lower hallucination rate compared to conventional Chain-of-Thought (CoT) approaches. Specifically, we show how the probabilistic formulation in Equation 4 systematically reduces the occurrence probabilities captured in Equations 2 and 3:
>     Let ε denote the hallucination probability at each reasoning step. We define the hallucination rates (aggregate probability of Equations 2 and 3) as:
>     - Chain-of-Thought (CoT):  $P_\mathrm{COT}({\text{Hal}}) = \epsilon$
>     - Our SQV framework: $P_\mathrm{SQV}({\text{Hal}}) = \epsilon^2$
>
>     **Theoretical Derivation**:
>     The SQV framework enables cross-path error cancellation through its adversarial structure. The correctness of final answers depends on three mutually exclusive scenarios:
>     1. Both Thesis and Antithesis incorrect: System likely produces incorrect output
>     2. One path correct, one incorrect: Verification mechanism weights correct path
>     3. Both paths correct: System guaranteed to produce correct output
>     Assuming the Verification phase correctly assesses statement reliability, and that Thesis (T) and Antithesis (A) errors are independent with $ε_T = ε_A = ε$, we derive:
>
>     $P_\mathrm{SQV}({\text{Hal}})$ = $P (\text{Hal}_T \cap \text{Hal}_A)$ = $\epsilon_T \cdot \epsilon_A = \epsilon ^2$
>
>     This establishes that $P_\mathrm{SQV}({\text{Hal}})$ = $\epsilon ^2 < \epsilon = P_\mathrm{COT}({\text{Hal}})$ for 0 < $\epsilon$ < 1.
>
>     Furthermore, we demonstrate $P_\mathrm{SQV}({\text{Hal}}) < P_\mathrm{SC}({\text{Hal}})$ when compared to Self-Consistency (SC) approaches. Detailed derivations are available in Appendix A.2.
>   - Ensemble Learning Interpretation (Appendix A.1)
>     We further analyze SQV through the lens of ensemble learning, providing insights into its effectiveness and suggesting directions for future research. The framework conceptualizes:
>     - Base learners: Thesis and Antithesis generators producing multiple reasoning paths
>     - Ensemble learner: Verification and Synthesis stages performing meta-reasoning
>     The counterfactual nature of SQV's base learners - specifically the opposing pair ($\theta_T$,$\theta_A$) - inherently promotes diversity among reasoning paths. This architectural characteristic aligns with established ensemble learning principles [10], where increased diversity among base learners leads to improved generalization performance.
>
>   Third, the core contribution extends beyond an "improved prompting technique." Our framework integrates two adaptation methods—zero-shot prompting (SQV_zero) and supervised fine-tuning (SQV_alignment)—to instill the SQV reasoning pattern in models. Prompting and SFT are methods for model adaptation; the core is the SQV reasoning pattern itself, whose performance warrant rigorous investigation. To the best of our knowledge, theoretical analyses of Chain-of-Thought mechanisms remain scarce [1], and our work represents an early but non-trivial step toward formalizing these processes.
>
>   In summary, the formalism is essential for making our method more interpretable. We hope this formalism provides a preliminary foundation that the community could find helpful for future exploration.

---

> > ### Comment · Reviewer_ckNu · 2025-11-26
> >
> > Thanks to the authors for the response.
> >
> > I looked at Appendix A and I find the "theoretical" analysis both vacuous and littered with unjustified assumptions and inconsistencies.
> >
> > 1. Both Eq. 4 and 12 both involve an expectation over all possible reasoning paths to present SQV as an ensemble learning method -- but in practice this is not done in the actual method. Self consistency uses multiple samples to approximate the marginalization over intermediate chain-of-thoughts, but I don't think SQV does this.
> >
> > 2. The crux of the argument for presenting SQV as a good ensemble learning is that it satisfies the diversity constraint KL(P_T||P_A) > \delta --> where does this come from? As far as i am aware, this is pulled out of thin air.
> >
> > 3. Eq. 9 for self-consistency is incorrect. If there is already marginalization over all possible reasoning steps T_k, what does the average over k represent?
> >
> > 4. For the hallucination analysis there are again unjustified assumptions that Hal_T and Hal_A are independent, and that \epsilon_T = \epsilon_A = \epsilon, and that \epsilon K < 1.
> >
> > Moreover, it is rather trivial to show that SQV is a special case of chain-of-thought. A model may decide to generate the thesis, antithesis, verification and synthesis steps on its own when doing chain-of-thought. Hence, any theoretical analysis which tries to show the superiority of SQV over the more general CoT is bound to be flawed.

---

> > > ### Author Response · Authors · 2025-11-27
> > > **Response to Reviewer ckNu [Further Questions]**
> > >
> > > Thanks for the response, we hereby address the concerns as follows:
> > >
> > > **Q1**: In Eq. 4 and 12, the expectation is not intended to signify an ensemble learning method, but rather to conceptualize the aggregation of all valid reasoning paths adhering to the SQV format (following the same expression manner in [1][2]). The core of ensemble learning lies in generating base learners and combining them into an ensemble predictor. To conceptualize the SQV process in ensemble learning manner, we proceed as follows: within SQV, **the (Thesis, Antithesis) stages can be viewed as producing two opposing base learners through a sequential generative process, while the (Verification, Synthesis) stages function as the ensemble learner**. It should be clarified that in SQV, a base learner refers to the pair ($\theta_T$,$\theta_A$), whereas in self-consistency, a base learner consists of $\theta_T$ by multiple sampling.
> > >
> > > **Q2**: This stems from the opposing nature between the two base learners, $\theta_T$ and $\theta_A$. As defined, $\theta_A$ is intended to challenge or question $\theta_T$.
> > >
> > > **Q3**: Thank you for your careful reading. The equation is correct but may require clarification. Here’s the intended explanation: The inner integral marginalizes over all possible instantiations represented as $T_k$ (a random variable) of the k-th reasoning path distribution, computing the predictive distribution for that specific base learner. The outer average aggregates across K distinct base learners, each corresponding to a different reasoning path distribution. The notation compactly represents a two-level ensemble: exact marginalization within each path's latent space, and empirical averaging across sampled paths.
> > >
> > > **Q4**: The independence assumption and the error term equality is a simplification for the situation, which establishes a baseline for the analysis. The Constraint εK < 1 aligns with real-world deployment trade-offs. For inference-time sampling in self-consistency, K is typically bounded to limit latency and compute costs.
> > >
> > > **Q5**: You're correct that it's technically possible for a model using general CoT to spontaneously generate steps resembling SQV's thesis, antithesis, verification, and synthesis. However, the core argument for SQV's empirical superiority isn't that this path is impossible in general CoT, but that it occurs with **vanishingly low probability without explicit adaption**. Faiththinker is a deliberate framework specifically proposed (via prompting or training adaptations) to significantly increase the likelihood that a model follows this highly structured  reasoning path.
> > >
> > > The theoretical analysis aims to demonstrate why this specific, enforced reasoning structure inherently leads to lower hallucination rates compared to the general CoT. The fact that SQV is a recoverable special case within CoT's expansive hypothesis space is precisely what makes it non-trivial: we are not analyzing arbitrary CoT trajectories, but rather the induced bias of a specific computational scaffold. The theoretical analysis demonstrates that when a model's reasoning is constrained to follow the thesis-antithesis-synthesis architecture—with explicit verification steps at critical junctures—the resulting error propagation dynamics differ qualitatively from unconstrained CoT.
> > >
> > > The empirical results in the paper– showing SQV consistently outperforming general CoT – directly support this. They suggest that while models might rarely stumble upon an SQV-like path spontaneously in general CoT, the probability is so low that it doesn't translate into reliable performance gains without the structured prompting or training provided by the SQV paradigm. Faiththinker effectively makes this robust reasoning pattern the default, high-probability path.
> > >
> > > **Reference**:
> > >
> > > [1] An Explanation of In-context Learning as Implicit Bayesian Inference
> > >
> > > [2] The Unified Cognitive Consciousness Theory for Language Models: Anchoring Semantics, Thresholds of Activation, and Emergent Reasoning

---

> ### Author Response · Authors · 2025-11-20
> **Response to Reviewer ckNu [Weakness 2-5]**
>
> - **W2**: The concepts of input-to-trajectory hallucination and intra-trajectory hallucination are discussed at length, but there is no empirical evidence about how prevalent these are, and whether the SQV technique actually reduces it. There is pretty much no analysis of what the SQV prompt does in practice for the problems studied here.
>
>  **A2**: To comprehensively assess the hallucination in reasoning steps, we expanded our evaluation framework beyond intrinsic metrics (top-10% group entropy) to include neural evaluation. Specifically, we employed LLM-as-a-Judge, a widely accepted approach for evaluating long-form reasoning quality [2][3][4]. We designed specialized prompts for DeepSeek-V3.1 to assess both input-to-trajectory hallucination (consistency with inputs) and intra-trajectory hallucination (internal logical coherence). For each sample, we conducted multiple independent evaluations and averaged the scores to ensure reliability. The results (in the updated pdf Figure 4) demonstrate that SQV reasoning significantly reduces both hallucination types compared to baseline chain-of-thought, yielding higher-quality reasoning trajectories.
>
> - **W3**: There are terms used throughout like top-10% group entropy, token efficiency without any explanation in the main paper. Looking at the appendix these are actually simple ideas (e.g., inverse of the response length) -- so why not directly use simple terminology?
>
> **A3**: For top-10% group entropy, we retained this term from [5] to measure reasoning trajectory quality, which maintains consistency with prior work and properly attribute the concept. We placed the detailed definition in the appendix to keep the main paper focused, but we understand this may hinder immediate comprehension.
>   For token efficiency, we centralized all metric definitions in the appendix's Metrics section for organizational uniformity, but we recognize this sacrifices clarity for brevity. Your suggestion is well-taken. We are happy to add brief parenthetical explanations in the revised manuscript.
>
> - **W4**: I find it very strange that the main table presents two metrics hallucination and accuracy, which always sum up to 1. There is no need to present two metrics -- this is like reporting both accuracy and error rates together.
>
>  **A4**: Thank you for the feedback. To improve clarity and presentation, we have updated Table 1 to include only accuracy rates.
>
> - **W5**: At a high-level, the paper fails to make it clear what should be the overall objective in cases where the RAG context is noisy. Should the model rely on its internal knowledge in this case, or should it identify the problems in the context and report it? What is being measured by the benchmarks used here?
>
>  **A5**: Thank you for this insightful comment, which addresses a critical conceptual foundation. The overall objective in noisy RAG contexts, as established in our work, is faithfulness to the provided context—not factual correctness based on world knowledge. This follows the taxonomy distinguishing factuality hallucinations (ungrounded content) from faithfulness hallucinations (discrepancy from context) [6][7][8].
>   To directly answer your question: the model should identify context problems and report them (via abstention) when the context is insufficient, and follow the context even when it conflicts with internal knowledge. These represent two facets of faithfulness:
>   1. Knowledge Conflict Resolution: When parametric knowledge contradicts the context (our counterfactual setting), faithfulness requires overriding internal knowledge to align with the context.
>   2. Abstention for Unanswerable Queries: When context is noisy or incomplete (our unanswerable setting), faithfulness requires recognizing deficiencies and abstaining rather than hallucinating.
>   Your scenario of "relying on internal knowledge or identifying context problems" specifically describes knowledge conflicts. We clarify this is a subset of the broader faithfulness problem; as formalized in [9], faithfulness encompasses both knowledge conflicts and abstention. Our benchmarks measure faithfulness by quantifying: (a) the rate at which models correctly prioritize context over parametric knowledge, and (b) the accuracy of abstention when context is inadequate. They do not evaluate factual accuracy against world knowledge.
>   Thus, the objective is clear: to evaluate whether models remain faithful to retrieved contexts, handling both conflicts and deficiencies appropriately.

---

> > ### Comment · Reviewer_ckNu · 2025-11-26
> >
> > 1. Thanks for including the new results -- please also include the prompts for the LLM-as-a-judge used to evaluate the hallucination types. Also some examples of these hallucination types in the appendix would be helpful.

---

> > > ### Author Response · Authors · 2025-11-27
> > > **Response to Reviewer ckNu [Further Questions]**
> > >
> > > **1**: LLM-as-a-judge Prompt:
> > >
> > > You are a professional reasoning quality assessment expert. Your task is to rigorously evaluate a given reasoning trajectory for hallucination issues across two distinct dimensions.
> > >  Evaluation Dimensions
> > > 1. Input-to-Trajectory Hallucination
> > > **Definition**: Assess whether statements, facts, or inferences in the reasoning trajectory contradict the provided context, or introduce unsupported information not present in the input.
> > > **Scoring Criteria (0-10)**:
> > > - **10**: Perfectly grounded in context. All claims have explicit support. Zero contradictions.
> > > - **7-9**: Mostly aligned with context. Minor generalizations or interpretative biases that don't affect core logic.
> > > - **4-6**: Noticeable inconsistencies with context, or 1-2 unsupported assumptions that impact reasoning.
> > > - **1-3**: Multiple contradictions or heavy reliance on unprovided information for key inferences.
> > > - **0**: Completely detached from context. Core facts severely conflict with input.
> > > **Checklist**:
> > > - Direct factual contradictions with context
> > > - Treating possibilities as certainties
> > > - Introducing unmentioned entities, events, or relationships
> > > - Misrepresenting or misreading key information from context
> > >  2. Intra-Trajectory Hallucination
> > > **Definition**: Evaluate the internal logical coherence of the trajectory, checking for self-contradictions, logical leaps, or causal inconsistencies within the reasoning chain.
> > > **Scoring Criteria (0-10)**:
> > > - **10**: Flawlessly coherent. Each step logically follows with no contradictions.
> > > - **7-9**: Generally sound. Minor inferential jumps or redundant steps only.
> > > - **4-6**: Obvious logical gaps or 1-2 self-contradictions that don't invalidate the final conclusion.
> > > - **1-3**: Multiple broken links or severe contradictions that undermine the reasoning.
> > > - **0**: Complete internal collapse. Self-contradictory throughout.
> > > **Checklist**:
> > > - Causal inversion or logical conflicts between steps
> > > - "Because A, therefore not A" type contradictions
> > > - Unbridgeable logical gaps between steps
> > > - Later steps undermining earlier premises
> > > - Using unproven intermediate conclusions as foundations
> > >
> > > Materials to Evaluate
> > > **Question**: {%s}
> > > **Context**: {%s}
> > > **Reasoning Trajectory**:
> > > {%s}
> > > Output Format
> > > Provide evaluation results in JSON with these fields:
> > > {
> > >  "input_to_trajectory_hallucination": {
> > >    "score": 0-10,
> > >    "analysis": "Detailed analysis of conflicts with context, citing specific text fragments",
> > >    "conflict_examples": [
> > >      {
> > >        "trajectory_claim": "Specific claim from trajectory",
> > >        "context_evidence": "Corresponding information from context",
> > >        "conflict_type": "Type: factual_conflict/over_inference/fabrication/misinterpretation"
> > >      }
> > >    ],
> > >
> > >  },
> > >  "intra_trajectory_hallucination": {
> > >    "score": 0-10,
> > >    "analysis": "Detailed analysis of internal logical breaks and contradictions",
> > >    "coherence_issues": [
> > >      {
> > >        "step_i": "Content of step i",
> > >        "step_j": "Content of step j",
> > >        "issue": "Description: self_contradiction/logical_leap/causal_inversion"
> > >      }
> > >    ],
> > >
> > >  },
> > > }
> > >
> > > **2**: We have added the examples in Appendix F in the updated pdf.

---

> ### Author Response · Authors · 2025-11-20
> **Response to Reviewer ckNu [Weakness 6 & References]**
>
> - **W6**: Some important citations are missing, e.g., an ICLR 2025 paper .
>
>  **A6**: Our work follows existing literature [6][7][8] in distinguishing between **factual consistency (alignment with world knowledge)** and **faithfulness (alignment with provided context)**. The cited ICLR 2025 paper focuses on situated faithfulness through dynamic balance between internal knowledge and external context to resolve knowledge conflicts, primarily addressing **factuality** problem. In contrast, our optimization objective specifically targets **context faithfulness** by minimizing hallucinations through strict adherence to given information. While both approaches address hallucination mitigation, the fundamental divergence lies in their operational definitions of fidelity – **theirs centers on factuality through knowledge reconciliation, whereas ours emphasizes contextual faithfulness through source-bound generation**. This distinction in core objectives resulted in limited perceived overlap during our literature review. We acknowledge this oversight and will incorporate appropriate citations in the revised manuscript to better position our work within the evolving research landscape.
>
> **References**:
>
> [1] A Survey of Frontiers in LLM Reasoning: Inference Scaling, Learning to Reason, and Agentic Systems (Ke et al.)
>
> [2] A Chain-of-Thought Is as Strong as Its Weakest Link: A Benchmark for Verifiers of Reasoning Chains (Jacovi et al., ACL 2024)
>
> [3] Synchronous Faithfulness Monitoring for Trustworthy Retrieval-Augmented Generation (Wu et al., EMNLP 2024)
>
> [4] RAGTruth: A Hallucination Corpus for Developing Trustworthy Retrieval-Augmented Language Models (Niu et al.)
>
> [5] Deep Think with Confidence (Fu et al.)
>
> [6] A Survey on Hallucination in Large Language Models: Principles, Taxonomy, Challenges, and Open Questions. (Huang et al., ACM Trans. Inf. Syst. 43, 2, Article 42 (March 2025), 55 pages)
>
> [7] Cognitive mirage: A review of hallucinations in large language models (Ye et al.)
>
> [8] FaithEval: Can Your Language Model Stay Faithful to Context, Even If "The Moon is Made of Marshmallows" (Ming et al., ICLR 2025)
>
> [9] Context-faithful Prompting for Large Language Models (Zhou et al., EMNLP 2023)
>
> [10] A survey of ensemble learning: Concepts, algorithms, applications, and prospects (Zhou, Z.H., et al., "IEEE Transactions on Knowledge and Data Engineering (2022)).

---

### Meta-Review · Area_Chair_NLSp · 2026-01-02

**Summary:**

This paper proposed FaithThinker, a new method to improve the faithfulness of large language models. The method designs a Self-Questioning and Verification strategy inspired by dialectical thinking. SQV-alignment is introduced as an adversarial context-augmented fine-tuning method that efficiently transfers SQV from large to small models. The experiments demonstrate the state-of-the-art robustness compared with CoT and Self-Refine.

The reviewers have mixed ratings for the paper. They acknowledged the novelty of the idea, the clear presentation and notable improvements. However, the reviewers have raised several major concerns about this paper.
- Unclear mathematical details: Reviewer ckNu pointed out that some math introduced for the technique are not very clear. The theoretical analysis is vacuous and littered with unjustified assumptions and inconsistencies.
- Unjustified concepts: Reviewer ckNu pointed out that some concepts (e.g., input-to-trajectory hallucination and intra-trajectory hallucination) are not justified by the proposed method.
- Reliance on teacher model: Reviewer u4ka pointed out that the method relies on a strong SQV teacher model, which is likely a kind of knowledge distillation.
- Lack of experiments on math reasoning: Reviewer u4ka and Reviewer JMUb pointed out the lack of experiments on math reasoning tasks.
- Computation efficiency: Reviewer 7zNZ and Reviewer JMUb asked the computation cost and efficiency compared with other fine-tuning methods.

**Reviewer Concerns:**

The authors have provided detailed rebuttal to address the reviewer concerns. But the reviewers did not actively engaged in the discussion. Reviewer ckNu had a follow-up comment regarding the unclear theoretical analysis. Based on the author rebuttal, I think some of the concerns have been addressed, including clarification on experiments on math reasoning, computation efficiency, and other individual comments. However, I think some of the concerns may not be addressed, including some unclear mathematical analysis (as the reviewer did not acknowledge that), and reliance on strong teacher model.

**Reviewer Scores:**

The reviewers did not acknowledge that they would change their score. The paper initially received mixing ratings (2,4,6,6). Since some of the concerns may not be addressed during rebuttal, I believe not all reviewers would change their score. As a result, I would recommend rejection.

---

### Decision · Program_Chairs · 2026-01-26

Reject